# HVR-Met: A Hypothesis-Verification-Replanning Agentic System for Extreme Weather Diagnosis

**Shuo Tang** [1 2 3]   **Jiadong Zhang** [1 2 3]   **Gengxian Zhou** [3 4]   **Qizhao Jin** [5]   **Qinxuan Wang** [1 2]   **Yi Hu** [5]   **Ning Hu** [5]
**Hongchang Ren** [5]   **Lingli He** [5]   **Shiming Xiang** [1 2]   **Jingtao Ding** [6]   **Jian Xu** [1 2]   **Jiaolan Fu** [5]   **Cheng-Lin Liu** [1 2 3]

## Abstract

While deep learning-based weather forecasting paradigms have made significant strides, addressing extreme weather diagnostics remains a formidable challenge. This gap exists primarily because the diagnostic process demands sophisticated multi-step logical reasoning, dynamic tool invocation, and expert-level prior judgment. Although agents possess inherent advantages in task decomposition and autonomous execution, current architectures are still hampered by critical bottlenecks: inadequate expert knowledge integration, a lack of professional-grade iterative reasoning loops, and the absence of fine-grained validation and evaluation systems for complex workflows under extreme conditions. To this end, we propose HVR-Met, a multi-agent meteorological diagnostic system characterized by the deep integration of expert knowledge. Its central innovation is the "Hypothesis-Verification-Replanning" closed-loop mechanism, which facilitates sophisticated iterative reasoning for anomalous meteorological signals during extreme weather events. To bridge gaps within existing evaluation frameworks, we further introduce a novel benchmark focused on atomic-level sub-tasks. Experimental evidence demonstrates that the system excels in complex diagnostic scenarios.

## 1. Introduction

Extreme weather diagnosis constitutes the systematic process of deciphering the underlying causes and evolutionary logic of severe atmospheric events through sophisticated reasoning and expert judgment (Weisman & Klemp, 1982; Abulimiti et al., 2023; Kerr et al., 2019). This process is fundamental to meteorological services and public safety. Given that these events are characterized by sudden emergence and swift development, operational centers must promptly identify and interpret anomalous signals (Brunet et al., 2023). Rapidly determining the specific nature and operational drivers of such events within constrained timeframes provides the essential scientific basis for weather alerts and strategic emergency responses (Zscheischler et al., 2020).

At present, AI cannot independently carry out operational extreme weather diagnosis (Yang et al., 2024a). In practice, this sophisticated workflow still depends on human forecasters, who must rapidly examine a small set of high-impact meteorological data and diagnostic indices under strict time constraints to isolate key drivers and craft an actionable interpretation (Heinselman et al., 2024). Despite its effectiveness, this manual pipeline is inherently constrained by strong dependence on individual experience, substantial labor demands, and heightened susceptibility to mistakes, particularly when less experienced personnel are involved (Doswell III, 2004). More fundamentally, extreme weather diagnosis requires analysts to pinpoint physically meaningful anomalies from fragmented, multivariate, and highly coupled datasets, then synthesize meteorological principles with situational context to perform disciplined attribution reasoning. Current deep-learning approaches typically fall short of this requirement: they struggle to consistently detect salient, event-specific signals across heterogeneous variables and scales, and they lack the ability to explicitly assemble a logically coherent and interpretable chain of evidence that supports a defensible warning decision.

Agentic frameworks offer considerable potential to address these operational limitations by automating specialized tasks (Wang et al., 2025) such as data acquisition (Jin et al., 2024; Qu et al., 2025; Lu & Wang, 2025; Schmidgall et al., 2025), code generation (Yang et al., 2024b; Zhang et al., 2025;

---

[1]MAIS, Institute of Automation, Chinese Academy of Sciences [2]School of Artificial Intelligence, University of Chinese Academy of Sciences [3]Zhongguancun Academy, Beijing [4]Beijing University of Posts and Telecommunications, Beijing [5]China Meteorological Administration [6]Tsinghua University, Beijing. Correspondence to: Jingtao Ding <dingjt15@tsinghua.org.cn>, Jian Xu <jian.xu@ia.ac.cn>, Jiaolan Fu <fujiaolan@cma.gov.cn>, Cheng-Lin Liu <liucl@nlpr.ia.ac.cn>.

Li et al., 2025), and multimodal interpretation (Bai et al., 2025a;b; Chen et al., 2025b). However, bridging the gap between general automation agents and professional-grade weather diagnosis agents necessitates overcoming two primary challenges. The first challenge is the insufficient integration of domain-specific expertise. Since the requirement for experienced operational expertise in selecting key diagnostic elements, general-purpose agents often prove incapable of autonomously identifying core evidence within complex meteorological scenarios. The second challenge lies in the lack of a heuristic reasoning loop for purposeful evidence collection. In extreme weather diagnosis, the discovery of one anomalous signal often dictates the specific tools or indices required for the next phase of analysis.

To address these two primary challenges, this paper proposes a multi-agent system specifically designed for the diagnostic analysis of extreme weather. Integrating seven functional agents such as Decomposer, Data Specialist, Code Executor, Plotter, Image Checker, Evaluators, Diagnostician and Reporters, the system enables automation spanning high-dimensional data retrieval, indicator computation, multimodal interpretation, and structured report generation. To tackle the first challenge, we established a diagnostic guideline repository as the core knowledge foundation. This repository was built by semi-automatically extracting key insights from 584 papers of extreme weather papers: Meteorological Monthly[1], Transactions of Atmospheric Sciences[2], Acta Meteorologica Sinica[3], American Meteorological Society[4] followed by rigorous validation by five senior forecasters to ensure professional-grade variable selection and index calculation. For the second challenge, the system introduces a "hypothesis-verification-replanning" reasoning mechanism that simulates the cognitive agentic workflow of human experts. Within this framework, the system formulates testable physical hypotheses based on anomalous signals and utilizes the guideline repository to plan diagnostic pathways. Should evidence be insufficient, the system dynamically triggers replanning to adjust its diagnostic path, thereby constructing a logically consistent, evidence-based diagnostic chain within a closed-loop reasoning process.

Furthermore, we present a comprehensive benchmark designed to evaluate both the reliability of individual operational units and the overall versatility of the agentic system. This benchmark comprises 100 extreme weather events for full-process diagnostic analysis, complemented by 250 specialized QA pairs for granular verification: 150 for meteorological index calculation (covering 30 index types) and 100 for diagnostic plotting (covering 20 figure categories).

---

[1]http://qxqk.nmc.cn/qxen/home
[2]http://dqkxxb.ijournals.cn/dqkxxben/home
[3]http://qxxb.cmsjournal.net/AboutJournal
[4]https://www.ametsoc.org/ams

All benchmark samples were reviewed and validated by experienced meteorologists. By covering a spectrum of tasks ranging from atomic operations, including standalone plotting and index calculation, to complex, holistic retrospective analysis, this benchmark provides rigorous validation of the system's robustness and versatility in real-world extreme weather diagnosis.

The contributions are as follows:

- We propose HVR-Met, a novel multi-agent framework designed to automate professional-grade extreme weather diagnosis.

- We introduce a Hypothesis–Verification–Replanning loop to self-improve the diagnostic pathway by focusing on anomalous meteorological signals.

- We construct a new benchmark for extreme weather diagnostics, covering 30 types of meteorological index calculation and 20 types of meteorological plotting.

- Our System achieves pass rates of 71.86% for index calculation, 79.52% for figure generation, and 85% for final reporting, demonstrating its robust capacity to assist forecasters in diagnosing extreme weather events.

## 2. Related Work

**Weather Foundation Models.** Deep learning-based weather forecasting systems (e.g., FourCastNet (Kurth et al., 2023), Pangu-Weather (Bi et al., 2023), NowcastNet (Zhang et al., 2023), GraphCast (Lam et al., 2023), FuXi (Chen et al., 2023), Stormer (Nguyen et al., 2024), GenCast (Price et al., 2025), FengWu (Chen et al., 2025a)) and weather foundation models (e.g., Climax (Nguyen et al., 2023), Aurora (Bodnar et al., 2025)), trained on large-scale structured numerical data, have significantly outperformed traditional physics-based numerical weather prediction (NWP) systems (Molteni et al., 1996) in terms of forecasting accuracy, computational efficiency, and task diversity. While these models have achieved major breakthroughs in numerical fidelity, they remain highly optimized "black-box" prediction tools. Due to the lack of explicit modeling of physical processes, these systems possess inherent limitations in causal reasoning and diagnostic interpretation, and they cannot support interactive scientific exploration or cross-domain reasoning through natural language interfaces.

**Meteorology Autonomous Agentic Frameworks.** In recent years, the focus of AI for meteorology has transferred from data-driven models to knowledge-driven autonomous agents (Guo et al., 2025; Liu et al., 2026; Pantiukhin et al., 2026). Zephyrus (Varambally et al., 2025) is the first agentic framework specifically designed for weather science,

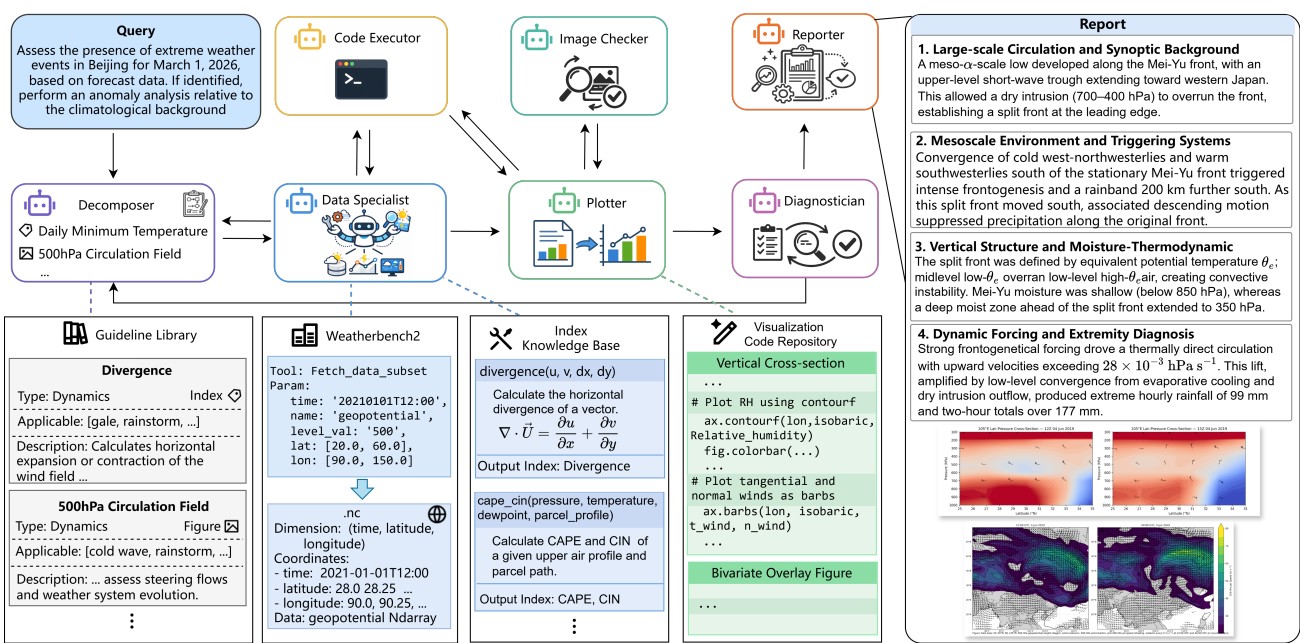

*Figure 1.* **Overview of the HVR-Met Framework.** Designed to emulate the professional "Weather Consultation" process, HVR-Met is a multi-agent system that automates extreme weather diagnosis through a dynamic *Hypothesis–Verification–Replanning* loop. The framework orchestrates seven specialized agents to collaboratively execute diagnostic tasks: the **Decomposer** for strategic planning, the **Data Specialist** and **Code Executor** for rigorous data retrieval and computation, the **Plotter** and **Image Checker** for standardized visualization and quality assurance, the **Diagnostician** for multi-modal abductive reasoning, and the **Reporter** for synthesizing the final diagnostic report.

constructing a meteorological agent environment and integrating meteorological tools. ClimateAgent (Kim et al., 2025) and EWE (Jiang et al., 2025) both introduced multi-agent systems; ClimateAgent is designed to address climate science tasks, while EWE utilizes predefined thought path for the retrospective analysis of extreme weather events.

Yet, the application of autonomous agentic system to the detection and causal attribution of extreme weather remains an unexplored frontier. Building upon these advances, we propose a novel agentic workflow for extreme weather diagnosis and analysis. Our Multi-Agent System identifies extreme weather types by detecting anomalous signals and iteratively self-optimizes its diagnostic pathways through continuous evaluation.

## 3. Methodology

### 3.1. Semi-automatic Construction of the Meteorological Knowledge Base

The Guideline Library characterizes the shared patterns of which meteorological indices and meteorological figures should be prioritized for different extreme weather types, and provides them as structured entries that the Decomposer can retrieve during task planning. As shown in Figure 1,

the Guideline Library encompasses information across four dimensions: the knowledge category, such as dynamics, thermodynamics, or moisture; the modality, either Index or Figure; the set of applicable weather types, including gale, rainstorm, cold wave, heat wave, and snowstorm; and a standardized description that explains the physical meaning of the index or figure and its diagnostic value.

To semi-automatically construct the Guideline Library, we collect 584 papers on extreme weather diagnosis. First, we employ MinerU (Niu et al., 2025) to parse each paper and extract structural elements. During this process, we focus exclusively on figure captions and the names of meteorological indices accompanied by explicit numeric values in the text, which allows us to identify the critical diagnostic figures and indices targeted for each extreme weather event. Next, we categorize the extracted records into five extreme weather types and consolidate identical figure and index names within each type. For the consolidated entries, we provide their standardized definitions and classify them into specific physical categories, including dynamics, thermodynamics, and moisture. Subsequently, five senior meteorological forecasters manually verify the library to ensure the accuracy of physical meanings and operational applicability. Finally, the structured Guideline Library can be directly invoked by the Decomposer, providing reliable

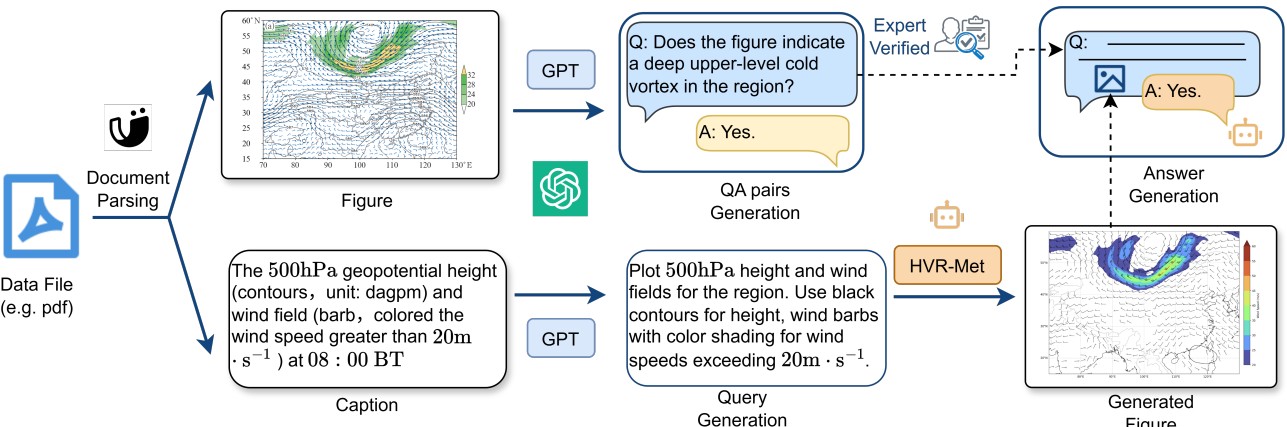

*Figure 2.* **The Verification Pipeline for Figure Generation.** We evaluate the "meteorological semantic integrity" of agent-generated visualizations via two parallel tracks: (1) **Ground Truth Construction (Top Branch):** "Gold standard" figures are extracted from extreme weather diagnosis papers, and a VLM generates binary QA pairs (e.g., checking for specific anomalies like vortices) which are rigorously verified by senior forecasters. (2) **Agent Evaluation (Bottom Branch):** Plotting requirements extracted from the original captions prompt the HVR-Met agent to autonomously generate and execute visualization code. Finally, the generated figure is fed back into the VLM to answer the original validation questions. The final score is quantified as the percentage of semantic alignment between the agent's output and the ground truth logic, ensuring physically consistent diagnostic visualization.

domain constraints and interpretable references for the HVR meteorological diagnosis.

In addition to the Guideline Library, we build an index knowledge base and a visualization code repository for executable analysis to support code generation during index calculation and figure plotting. Specifically, we automatically crawl and organize the documentation of index calculation functions in MetPy that are relevant to meteorological diagnosis, with emphasis on required input variables, unit constraints, output definitions, and typical usage examples. In parallel, we construct a visualization code repository by collecting professional meteorological plotting scripts based on Cartopy and Matplotlib. This repository covers core operations needed for operational meteorological plotting, including map projections, geographic feature overlays, contour drawing, colorbars, and annotations. To equip the agents with domain-specific expertise, the index knowledge base and the visualization code repository are attached as RAG modules to the Data Specialist and the Plotter, respectively. This enables them to retrieve necessary knowledge when generating index calculation and plotting codes, thereby reducing execution failures and result deviations caused by parameter misuse, unit inconsistencies, and missing plotting steps.

### 3.2. Multi-Agent Diagnostic Framework

The multifaceted nature of extreme weather diagnosis, which requires the integration of high-dimensional data retrieval, rigorous index calculation, and synoptic reasoning, poses significant challenges for LLMs. To address this,

we introduce HVR-Met, a multi-agent meteorological diagnostic framework. Designed to emulate the professional "Weather Consultation" workflow, this system establishes a collaborative agentic environment that unifies domain-specific capabilities through structured role specialization.

The framework encompasses seven specialized agents:

**Decomposer**: Serves as the strategic planner that parses high-level diagnostic queries. Drawing on the historical experience from the Guideline Library, it formulates an initial hypothesis regarding which meteorological figures or indices are likely to exhibit anomalies, and subsequently decomposes the problem into a logical sequence of executable sub-tasks for downstream agents.

**Data Specialist**: Serves as the data-retrieval and index calculation module. It references the Index Knowledge Base to synthesize Python scripts for accessing high-dimensional meteorological tensors from WeatherBench2 (Rasp et al., 2024) and generating index calculation code, ensuring rigorous preprocessing and numerical precision.

**Code Executor**: Serves as a sandboxed execution environment for securely running generated scripts. It manages dependencies and captures execution feedback, maintaining a robust separation between code synthesis and runtime behavior.

**Plotter**: Serves as the synoptic visualization module. Integrated with the visualization code repository, it generates code to render meteorological fields into high-quality figures complying with standard meteorological plotting specifications, leveraging appropriate geospatial projections and col-

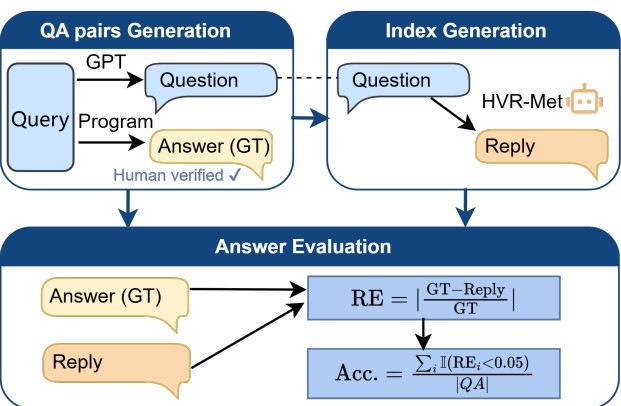

*Figure 3.* **The Evaluation Pipeline for Meteorological index calculation.** This framework quantifies the agent's numerical precision against human-verified standards. The workflow proceeds in three stages: (1) **Ground Truth Construction (Top-Left):** Situational questions are formulated via LLMs (GPT), while the ground-truth values (GT) are derived from raw data using expert-grade programs. (2) **Agent Inference (Top-Right):** The HVR-Met agent processes the question to compute a predicted index value (labeled as 'Reply'). (3) **Metric Calculation (Bottom):** The system evaluates accuracy by calculating the Relative Error (RE) between the Reply and GT. A prediction is accepted as correct only if the absolute relative error is strictly below 0.05.

ormaps to translate numerical data into interpretable visual evidence.

**Image Checker**: Serves as the quality-assurance module. It validates generated figures against established meteorological plotting standards (e.g., contour intervals and label placement) to ensure clarity and domain compliance prior to analysis.

**Diagnostician**: Serves as the core inference unit that approximates expert reasoning. It first evaluates the initially hypothesized figures and indices. If the anticipated anomalies are absent, it triggers a Hypothesis-Verification-Refinement (HVR) loop to adjust the diagnostic direction. Ultimately, by integrating the validated multimodal evidence with domain knowledge, it performs abductive reasoning to identify the root causes driving the extreme weather event.

**Reporter**: Serves as the reporting module that consolidates diagnostic outputs. It organizes the reasoning trace and supporting evidence into a comprehensive diagnostic report.

### 3.3. Hypothesis–Verification–Replanning

To emulate the cognitive rigor of human forecasters, we design a dynamic inference loop capable of self-correction. The mechanism unfolds in four sequential phases:

**Event-Driven Hypothesis Generation.** Upon receiving a query, the Decomposer first instructs the Data Specialist to retrieve the forecast data and evaluate it against standardized

meteorological thresholds. The Data Specialist accurately classifies the target event into one of five predefined categories, namely rainstorm, snowstorm, gale, heat wave, or cold wave, and returns this classification to the Decomposer. Subsequently, the Decomposer formulates an initial hypothesis by selecting five relevant diagnostic factors, which comprise specific meteorological indices and figure types, from the Guideline Library, thereby constructing an executable task chain.

**Multi-Modal Verification Execution.** Following the task chain, the Data Specialist computes the selected indices, while the Plotter renders the corresponding meteorological figures. To ensure compliance with operational meteorological plotting standards, all generated figures must pass validation by the Image Checker. This step effectively translates the abstract diagnostic hypothesis into intuitive, multimodal evidence.

**Discrepancy Detection and Evidence Inference.** The Diagnostician evaluates the multimodal evidence to determine whether the hypothesized anomalies are present. An "anomaly" is strictly defined as an obvious manifestation of extreme weather. For figures, this requires the presence of distinct visual synoptic features, such as vortices, shear lines, or cold fronts. For indices, it requires the computed values to deviate significantly from the 20-year climatological mean.

**Feedback-Driven Replanning.** The system requires a set of five confirmed anomalous factors to provide sufficient evidence for a definitive diagnosis. If all selected factors exhibit the anticipated anomalies, the Reporter aggregates the findings and generates the final diagnostic report. However, if any factor fails to show the expected anomaly, the system records these negative results in its short-term memory, retrieves alternative causal logic from the Guideline Library, and initiates a new verification iteration. This dynamic HVR loop continues until five anomalous factors are successfully confirmed or a maximum limit of 50 turns is reached, at which point the Reporter concludes the diagnosis.

## 4. A Comprehensive Benchmark for Extreme Weather Diagnosis

We present a comprehensive benchmark designed to evaluate both the reliability of individual operational units and the overall versatility of the agentic system. Specifically, this benchmark is structured across three distinct evaluation components. These include the full-pipeline diagnostic workflow, which executes the complete extreme weather diagnosis across all seven agents, as well as two pivotal sub-tasks dedicated to isolated figure generation and index calculation.

*Table 1.* Performance evaluation of different models across the five stages of the agentic workflow. Scores reflect the mean performance across 100 extreme weather cases (20 cases per category for gale, rainstorm, snowstorm, cold wave, and heat wave) based on our fine-grained scoring dimensions.

| MODEL | HYPOTHESIS | DATA | INDEX | FIGURE | FINAL REPORT |
|---|---|---|---|---|---|
| GPT-5 | 4.70 | 4.92 | 4.07 | 4.13 | 3.94 |
| GEMINI-3-PRO-PROVIEW-THINKING | 4.63 | 4.87 | 4.02 | 3.98 | 3.76 |
| DEEPSEEK-R1-0528 | 3.91 | 4.25 | 3.56 | 3.41 | 2.97 |
| QWEN3-CODER-480B-A35B-INSTRUCT | 3.73 | 4.31 | 3.30 | 2.46 | 2.33 |

### 4.1. Full-Pipeline Diagnostic Workflow

For full-pipeline diagnostic workflow, we rigorously curated a dataset of 100 distinct extreme weather events, comprising exactly 20 samples for each of the five types: rainstorm, snowstorm, gale, heat wave, and cold wave. To ensure fairness, all these selected events postdate the knowledge cutoff dates of the underlying models driving the agents. Crucially, the data source for this evaluation is constructed independently from the corpus used for the subsequent sub-task benchmarks. This strict data isolation ensures that the agent's holistic reasoning capabilities are evaluated on entirely unseen scenarios, effectively preventing any potential data leakage.

To evaluate the quality of the comprehensive diagnostic reports, we collaborated with five senior forecasters to formulate a professional-grade LLM-based scoring rubric (detailed in the Appendix). Specifically, this assessment framework employs a standardized 0–5 scoring scale across all dimensions, where a score greater than 4 is considered a pass.

Furthermore, to ensure the reliability of this automated evaluation, we conducted a human evaluation by randomly sampling 30% of the dataset (30 events) for manual scoring by domain experts. The high correlation observed between the expert scores and the LLM evaluations validates the operational accuracy of our automated scoring workflow.

### 4.2. The Specific Sub-task Evaluation

For the specific sub-task evaluation, we present a multifaceted semi-synthetic benchmark derived from 584 extreme weather analysis papers published in prestigious journals, including Weather and Forecasting (AMS), Acta Meteorologica Sinica, and Meteorology. This benchmark is structured to evaluate the multi-agent system across two core functional dimensions: figure generation and meteorological index calculation.

The first dimension evaluates the accuracy of the meteorological figures generated by the agent. This track features 100 high-quality tasks across 20 core figure types. Unlike conventional pixel-level similarity metrics, we propose an assessment agentic workflow centered on "meteorological

semantic integrity," which prioritizes whether the agent-generated code produces visualizations that accurately represent critical atmospheric anomalies such as vortices and shear lines. To implement this, we select 100 "gold standard" figures from the 584 papers and employ a VLM to generate binary yes-or-no QA pairs based on these original figures. These pairs are subsequently rigorously verified by five senior forecasters to ensure their operational relevance, as well as the accuracy and uniqueness of the answers.

During evaluation, the agent is tasked with autonomously generating a meteorological figure that matches the exact timestamp, spatial coordinates, and variables of the corresponding gold standard figure. After the agent executes its code to produce the visualization, this generated figure is fed into the VLM alongside the original question from the human verified question and answer pair. The VLM then outputs a binary yes or no answer. By comparing the response of the VLM with the ground truth answer, we evaluate the capability of the agent to translate textual requirements into accurate visual representations.

Specifically, we quantify this capability using the mean accuracy across all validation queries. This overall average directly reflects whether the visualizations generated by the agent accurately capture the essential meteorological features present in the original figures. For the complete set of $N$ binary classification questions, the final visualization score is calculated as the average accuracy $\frac{1}{N} \sum_{i=1}^{N} \mathbb{I}(A_{gen}^{(i)} = A_{gt}^{(i)}) \times 100\%$. Here, $A_{gen}$ and $A_{gt}$ denote the answers derived by the VLM for the generated and original figures, respectively, while $\mathbb{I}(\cdot)$ is the indicator function for an exact match. The full workflow of the figure generation sub-task is as shown in Figure 2.

The second dimension evaluates the precision of the system in meteorological index calculation. This benchmark encompasses 150 tasks covering 30 essential indices. For each index, five situational questions are formulated via LLMs, with ground truth answers extracted from raw data using human verified, expert grade computational scripts to ensure high fidelity evaluation. The precision and robustness of the outputs generated by the agent are quantified through error analysis against these reference values. To rigorously quantify the computation accuracy, we adopt a strict acceptance

*Table 2.* Ablation study of the domain-specific knowledge bases using GPT-5. The framework's performance is evaluated across the five diagnostic stages by individually removing the Guideline Library, the Index Knowledge Base, and the Figure Knowledge Base.

| SETTING | HYPOTHESIS | DATA | INDEX | FIGURE | FINAL REPORT |
|---|---|---|---|---|---|
| w/o Guideline Library | 1.27 | 4.76 | 2.17 | 3.42 | 0.40 |
| w/o Index Knowledge Base | 3.97 | 4.64 | 3.23 | 4.08 | 1.57 |
| w/o Figure Knowledge Base | 3.90 | 4.69 | 4.01 | 3.88 | 3.45 |
| **Full Framework (Ours)** | **4.70** | **4.92** | **4.07** | **4.13** | **3.94** |

threshold: a prediction $Reply$ is considered correct only if the absolute relative error satisfies $\left|\frac{GT-Reply}{GT}\right| < 0.05$. For cases where the ground truth is $GT = 0$, we impose an absolute error constraint of $|Reply| < 0.05$ to ensure numerical stability. The final overall accuracy for this sub-task is then calculated as the percentage of all 150 computational queries that successfully meet these rigorous error conditions. The full workflow of the index calculation sub-task is as shown in Figure 3.

## 5. Experiment

### 5.1. Experiment Settings

We developed a multi-agent system for extreme weather diagnosis based on the AG2 (Wang et al., 2024) framework, selecting 100 cases across five typical categories including gale, rainstorm, snowstorm, cold wave, and heat wave, with 20 cases per category for our evaluation dataset. To systematically characterize the performance of the multi-agent workflow in complex diagnostic tasks, we designed five fine-grained scoring dimensions centered on critical intermediate stages: Hypothesis, Data, Index, Figure, and Report. This framework enables a granular assessment of the system's capabilities, encompassing diagnostic hypothesis and tool selection, data acquisition and preprocessing, meteorological index calculation, visualization, and the synthesis of comprehensive diagnostic reports.

### 5.2. Performance Comparison across Diagnostic Workflow Stages

As illustrated in Table 1, GPT-5 and Gemini-3-Pro-Preview-Thinking exhibit the best performance across all five evaluation dimensions. GPT-5 sets the state-of-the-art benchmark, highlighted by its peak Data (4.92) and Hypothesis (4.70) scores. The strong performance of both models in the initial Hypothesis phase (4.70 and 4.63, respectively) demonstrates their extensive meteorological prior knowledge and high-level strategic planning. This foundation enables them to accurately anchor the causal pathways for extreme weather diagnosis, complete HVR-loop iterations within just three rounds, and maintain robust output quality through to the Final Report stage (3.94 and 3.76).

Conversely, Deepseek-R1 and Qwen3-Coder experience

significant performance bottlenecks in the more complex, later diagnostic phases. Qwen3-Coder, for instance, sees a precipitous decline in the Figure and Final Report stages, dropping to scores of 2.46 and 2.33. Execution logs reveal that this degradation is primarily driven by an inability to handle dimension and unit conversions. When calling specialized meteorological libraries (e.g., MetPy, Cartopy), these models lack strict tool adherence and error-correction resilience; after only one or two trial-and-error attempts to fix parameters, they retreat to generic coding patterns, ultimately leading to computational failures and poor final deliverables.

To ensure the reliability of the aforementioned results, we report the Pearson correlation coefficients across all five scoring dimensions computed from a subset of $N = 30$ human-annotated cases. As further detailed in the Appendix, every dimension achieves $r \geq 0.81$ with $p \ll 0.01$. This confirms a strong positive correlation between the expert manual scores and the automated evaluations, thereby validating the accuracy and reliability of our current scoring workflow.

### 5.3. The Results of Atomic-level sub-tasks

To provide a more granular assessment of the multi-agent system's diagnostic capabilities in extreme weather scenarios, we categorize the sub-tasks into three levels of difficulty. For the figure generation task, the difficulty level is determined by the complexity of the layer overlays. Plotting a single meteorological field such as a 500 hPa geopotential height map is classified as Easy. In contrast, superimposing multiple fields such as overlaying a wind field onto the 500 hPa map elevates the task to Medium. For the index calculation task, the evaluation is stratified into basic statistical operations like finding extrema and means, single-formula calculations, and multi-step composite derivations. This evaluation approach is designed to probe the system's performance boundaries when confronted with varying degrees of visual and computational logic complexity.

Figure 4 presents an evaluation of computational accuracy across three difficulty levels, revealing an inverse relationship between task difficulty and model performance that underscores a significant reasoning bottleneck in current architectures. In the index calculation sub-task (Figure 4a),

*Table 3.* Ablation study of the multi-agent system components using GPT-5. We evaluate the contribution of the Decomposer, Image Checker, and Diagnostician modules across the five research stages. The results demonstrate the necessity of each component in maintaining the integrity of the full meteorological diagnostic workflow.

| GPT-5 | | | HYPOTHESIS | DATA | INDEX | FIGURE | FINAL REPORT |
|---|---|---|---|---|---|---|---|
| DECOMPOSER | IMAGE CHECKER | DIAGNOSIS | | | | | |
| × | √ | √ | 3.23 | 2.20 | 3.19 | 1.03 | 1.12 |
| √ | × | √ | 4.67 | 4.90 | 4.01 | 2.98 | 3.20 |
| √ | √ | × | 2.71 | 4.89 | 4.05 | 4.11 | 3.44 |
| √ | √ | √ | 4.70 | 4.92 | 4.07 | 4.13 | 3.94 |

reasoning-enhanced models demonstrate superior resilience on Hard-level indices requiring multi-variable retrieval and multi-step derivations. Specifically, Gemini3-Pro-thinking and DeepSeek-R1 secure 46.43% and 42.31% accuracy respectively, surpassing GPT-5's 31.25% . They achieve this robust performance by accurately decomposing complex formulas and iteratively calling numerical libraries (e.g., MetPy) to avoid error accumulation. In contrast, code-specialized models like Qwen3-Coder experience a precipitous decline to 26.32% at the Hard level. Log analysis reveals that their primary failures stem from dimensional handling errors during complex unit conversions. These failures highlight that maintaining precision across highly coupled meteorological reasoning pathways requires a deep understanding of meteorological principles beyond generalized coding skills.

Similarly, in the figure extraction sub-task (Figure 4b), plotting accuracy declines sharply with increased difficulty, evidenced by GPT-5 dropping from 96.31% on Easy tasks to 54.66% on Hard tasks. In these complex scenarios requiring multi-layer overlays, most errors are not numerical, but rather visual occlusions. Improper management of the multi-layer rendering order (z-order) obscures critical information, ultimately rendering the generated plots ineffective for professional meteorological analysis.

### 5.4. Ablation Study

To rigorously validate the necessity and individual contribution of each core component within our multi-agent framework, we conduct a series of ablation experiments focusing on the Decomposer, Image Checker, and Diagnosis modules.

Table 3 validates the critical role of each agent. Removing the Decomposer causes a systemic collapse, with the Final Report score plummeting to 1.12. This confirms its foundational role; without structured planning, downstream execution becomes incoherent. Ablating the Image Checker specifically degrades visual quality Figure score drops from 4.13 to 2.98, highlighting its necessity for ensuring meteorological plotting standards. The absence of the Diagnostician compromises the professional relevance of the

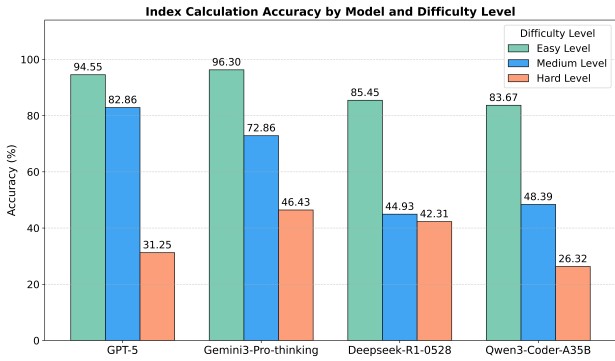

*a* Index calculation accuracy.

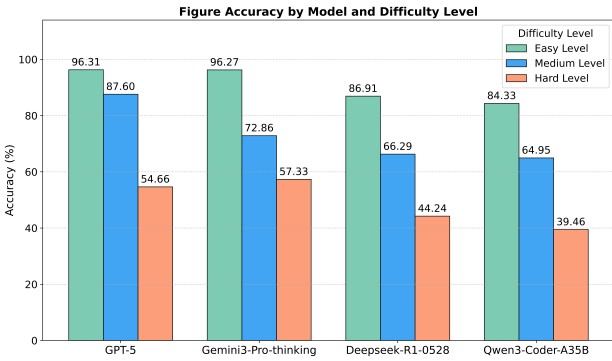

*b* Figure accuracy.

*Figure 4.* **Performance Evaluation by Task Type.** Comparison of model accuracy on (a) Index Calculation tasks and (b) Figure Extraction tasks.

autonomously selected figures and indices, leading to a consequent degradation in the final report quality.

Table 2 is the result of ablation study of the Guideline Library.The removal of the Guideline Library results in a precipitous decline in both Hypothesis (1.27) and Final Report (0.40) scores. This indicates that without the domain priors provided by the guidelines, the agent fails to lock onto critical meteorological variables, regressing from an expert system to a general-purpose language model with limited diagnostic capability. Furthermore, every diagnostic template within the library strictly corresponds to specific

executable entries in the Index and Figure Knowledge Bases. Consequently, the absence of the Guideline Library causes a cascading degradation in the subsequent index calculation and figure generation phases. Even though the downstream Knowledge Bases remain active, the agent loses the strategic direction required to retrieve and deploy the correct tools.

The removal of the Index or Figure Knowledge Bases leads to a degradation not only in index calculation and figure generation, but also notably in the Hypothesis score. This counter-intuitive decline stems from error propagation within the verification loop: the Diagnosis Agent attempts to validate hypotheses against erroneous indices and distorted figures. Due to this flawed evidence, the agent creates a mismatch between theoretical expectations and observed reality, leading it to incorrectly reject valid hypotheses (False Negatives) or accept invalid ones. This verification failure triggers unnecessary or misguided Replanning, forcing the agent to discard the correct initial hypothesis in favor of erroneous alternatives, ultimately compromising the final diagnostic accuracy.

The Data metric demonstrates remarkable stability across all ablation settings, maintaining high scores within the range of 4.64–4.92, indicating that the data retrieval capability is largely unaffected by the Knowledge Base configuration.

## 6. Limitations

A primary limitation of HVR-Met is its inherent dependence on input data quality. Our current evaluation utilizes reanalysis data to largely eliminate the errors present in raw forecasts. However, real-world operational applications must ingest live prediction data from various numerical models such as the CMA GRAPES model, the NOAA GFS model, and the ECMWF model. These operational models inevitably contain inherent computational biases and errors that can vary significantly even for the same weather event. When integrated into the diagnostic workflow, these primary data inaccuracies could cascade through the system and lead to the omission or misjudgment of critical weather features. Overcoming this operational bottleneck will ultimately require a collaborative approach where experienced forecasters synthesize the automated system outputs with localized observational data.

## 7. Conclusion

In this paper, we propose HVR-Met, a novel multi-agent framework designed to automate extreme weather diagnosis. Central to this system is the Hypothesis-Verification-Replanning loop. This iterative mechanism self-improves the diagnostic pathway by focusing on anomalous meteorological signals. By introducing expert knowledge into the system, this mechanism ensures that the automated workflow is driven by rigorous meteorological reasoning. Furthermore, we address the critical lack of evaluation frameworks for meteorological sub-tasks by constructing a new fine-grained benchmark. This comprehensive benchmark covers 30 types of meteorological index calculations and 20 types of meteorological plotting. Our experimental results highlight the efficacy of the system, featuring an 85% pass rate for final diagnostic reporting. The framework also delivers high performance in essential atomic tasks, achieving pass rates of 71.86% for index calculation and 79.52% for figure generation. These findings demonstrate that HVR-Met is a robust multi-agent system that has reached the diagnostic proficiency of a junior forecaster, proving its capability to effectively assist human experts in diagnosing extreme weather events.

## Acknowledgements

This work is sponsored by the Strategic Priority Research Program of the Chinese Academy of Sciences (No. XDA0480000), the Zhongguancun Academy Project No. 02012501 and Project No. C20250403, and the National Natural Science Foundation of China (U24B20180, 62476152).

## Impact Statement

This paper advances the field of Machine Learning by introducing a multi-agent framework capable of professional-grade logical reasoning in complex, high-stakes domains. The societal consequences of HVR-Met are particularly significant in the context of global climate change and the increasing frequency of extreme weather events

We do not foresee any significant negative ethical implications, as the system is designed to assist and augment—not replace—human expertise in protecting the public.

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

# A. Appendix

*Table 4.* Mean Relative Error of Index Calculation Results with Difficulty Classification.

| Index | Difficulty | GPT-5 | Gemini3-Pro-thinking | Deepseek-R1-0528 | Qwen3-Coder-A35B |
|---|---|---|---|---|---|
| *Easy Level* | | | | | |
| Cold High Pressure Intensity | Easy | $2.341 \times 10^{-6}$ | $3.860 \times 10^{-7}$ | $5.229 \times 10^{-4}$ | $7.459 \times 10^{-4}$ |
| Temperature | Easy | $3.842 \times 10^{-5}$ | $7.693 \times 10^{-5}$ | $5.964 \times 10^{-1}$ | $< 10^{-8}$ |
| Specific Humidity | Easy | $1.614 \times 10^{-2}$ | $1.603 \times 10^{-2}$ | $2.237 \times 10^{-2}$ | $1.615 \times 10^{-2}$ |
| Precipitable Water (PWAT) | Easy | $4.947 \times 10^{-4}$ | $3.638 \times 10^{-2}$ | $1.028 \times 10^{-1}$ | $1.708 \times 10^{-4}$ |
| 500hPa Geopotential Height | Easy | $2.372 \times 10^{-3}$ | $5.134 \times 10^{-7}$ | $3.728 \times 10^{-5}$ | $1.003 \times 10^{-6}$ |
| Surface Low-Pressure | Easy | $4.238 \times 10^{-5}$ | $5.717 \times 10^{-5}$ | $6.720 \times 10^{-3}$ | $1.482 \times 10^{-2}$ |
| Thunderstorm High Central Intensity | Easy | $1.585 \times 10^{-6}$ | $< 10^{-8}$ | $1.393 \times 10^{-5}$ | $< 10^{-8}$ |
| Cold Pool Central Temperature | Easy | $6.340 \times 10^{-3}$ | $1.108 \times 10^{-4}$ | $4.417 \times 10^{-3}$ | $6.317 \times 10^{-2}$ |
| Surface Wind Speed | Easy | $1.000 \times 10^{-2}$ | $< 10^{-8}$ | $< 10^{-8}$ | $3.250 \times 10^{-2}$ |
| 24-h Temp Change at Different Levels | Easy | $8.793 \times 10^{-5}$ | $4.085 \times 10^{-4}$ | $1.166 \times 10^{-4}$ | $3.711 \times 10^{-4}$ |
| Polar Vortex Center Geopotential Height | Easy | $4.979 \times 10^{-5}$ | $5.111 \times 10^{-5}$ | $2.227 \times 10^{-4}$ | $5.221 \times 10^{-5}$ |
| *Medium Level* | | | | | |
| Surface Negative Temp Advection | Medium | $3.429 \times 10^{-2}$ | $1.014 \times 10^{-1}$ | $1.366 \times 10^{-1}$ | $4.489 \times 10^{-1}$ |
| Positive Vorticity | Medium | $1.226 \times 10^{-2}$ | $2.343 \times 10^{-2}$ | $3.011 \times 10^{-2}$ | $8.187 \times 10^{-3}$ |
| Jet Intensity | Medium | $9.325 \times 10^{-5}$ | $3.359 \times 10^{-3}$ | $2.117 \times 10^{-2}$ | $3.775 \times 10^{-6}$ |
| Horizontal Temperature Gradient | Medium | $5.951 \times 10^{-4}$ | $2.078 \times 10^{-5}$ | $4.977 \times 10^{-2}$ | $1.208 \times 10^{-2}$ |
| Maximum Vertical Velocity | Medium | $7.989 \times 10^{-2}$ | $7.509 \times 10^{-2}$ | $7.511 \times 10^{-2}$ | $3.666 \times 10^{-1}$ |
| Low-Level Divergence Extrema | Medium | $3.252 \times 10^{-2}$ | $5.996 \times 10^{-2}$ | $8.198 \times 10^{-2}$ | $1.187 \times 10^{-1}$ |
| Warm Advection Center Intensity | Medium | $3.944 \times 10^{-4}$ | $1.026 \times 10^{-2}$ | $3.693 \times 10^{-2}$ | $3.389 \times 10^{-2}$ |
| Average Relative Humidity | Medium | $1.226 \times 10^{-2}$ | $3.235 \times 10^{-3}$ | $4.349 \times 10^{-3}$ | $1.355 \times 10^{-2}$ |
| High-Level Convergence Extrema | Medium | $2.997 \times 10^{-8}$ | $6.425 \times 10^{-2}$ | $9.891 \times 10^{-2}$ | $7.218 \times 10^{-2}$ |
| Surface Cyclone Pressure Change Rate | Medium | $4.845 \times 10^{-2}$ | $2.305 \times 10^{-3}$ | $5.728 \times 10^{-1}$ | $9.809 \times 10^{-2}$ |
| Equiv. Potential Temp Diff (850-500hPa) | Medium | $2.469 \times 10^{-4}$ | $1.311 \times 10^{-4}$ | $5.609 \times 10^{-2}$ | $5.655 \times 10^{-4}$ |
| 0°C Isotherm Height | Medium | $8.827 \times 10^{-4}$ | $5.154 \times 10^{-4}$ | $8.902 \times 10^{-3}$ | $1.875 \times 10^{-2}$ |
| Water Vapor Flux Convergence Intensity | Medium | $1.710 \times 10^{-3}$ | $4.394 \times 10^{-2}$ | $1.932 \times 10^{-1}$ | $8.804 \times 10^{-3}$ |
| Temp Standardized Anomaly (SA) | Medium | $2.970 \times 10^{-3}$ | $2.553 \times 10^{-3}$ | $6.998 \times 10^{-1}$ | $6.765 \times 10^{-1}$ |
| *Hard Level* | | | | | |
| Frontogenesis Function Center Value | Hard | $6.480 \times 10^{-2}$ | $2.295 \times 10^{-2}$ | $5.723 \times 10^{-2}$ | $1.673 \times 10^{-1}$ |
| Moisture Flux Divergence | Hard | $3.479 \times 10^{-3}$ | $9.846 \times 10^{-3}$ | $2.863 \times 10^{-1}$ | $1.387 \times 10^{-1}$ |
| CAPE | Hard | $1.625 \times 10^{-1}$ | $1.527 \times 10^{-1}$ | $6.898 \times 10^{-1}$ | $2.429 \times 10^{-1}$ |
| Vertical Wind Shear | Hard | $4.057 \times 10^{-2}$ | $3.559 \times 10^{-2}$ | $9.057 \times 10^{-1}$ | $4.961 \times 10^{-2}$ |
| 24-h Pressure Change Difference | Hard | $5.571 \times 10^{-1}$ | $1.370 \times 10^{-4}$ | $4.500 \times 10^{-5}$ | $1.047 \times 10^{-1}$ |

*Table 5.* Correlation between the automatic evaluation and human evaluation using the Pearson correlation coefficient.

| DIMENSION | $r$ | $p$-VALUE |
|---|---|---|
| HYPOTHESIS | 0.83 | $5.39 \times 10^{-27}$ |
| DATA | 0.94 | $9.67 \times 10^{-47}$ |
| INDEX | 0.98 | $1.08 \times 10^{-73}$ |
| FIGURE | 0.82 | $5.29 \times 10^{-25}$ |
| FINAL REPORT | 0.81 | $2.62 \times 10^{-24}$ |
| OVERALL (AVG.) | 0.88 | $< 10^{-24}$ |

## Index Knowledge Base Example

### metpy.calc.precipitable_water

```
metpy.calc.precipitable_water(pressure, dewpoint, *,
bottom=None, top=None)
```

Calculate precipitable water through the depth of a sounding. The formula used is:

$$-\frac{1}{\rho_l g} \int_{p_{\text{bottom}}}^{p_{\text{top}}} r \, dp$$

from [Salby1996], p. 28.

**Parameters:**

- **pressure** (*pint.Quantity*) – Atmospheric pressure profile.

- **dewpoint** (*pint.Quantity*) – Atmospheric dewpoint profile.

- **bottom** (*pint.Quantity, optional*) – Bottom of the layer, specified in pressure. Defaults to `None` (highest pressure).

- **top** (*pint.Quantity, optional*) – Top of the layer, specified in pressure. Defaults to `None` (lowest pressure).

**Returns:**
  *pint.Quantity* – Precipitable water in the layer.

**Examples:**

```
>>> pressure = np.array([1000, 950, 900]) * units.hPa
>>> dewpoint = np.array([20, 15, 10]) * units.degC
>>> pw = precipitable_water(pressure, dewpoint)
```

**Notes:**

- Only functions on 1D profiles (not higher-dimension vertical cross sections or grids).

- *Changed in version 1.0:* Signature changed from `(dewpt, pressure, bottom=None, top=None)`.

*Figure 5.* Guide Library Example: metpy.calc.precipitable_water.

## Decomposer System Message

You are the **Lead Meteorological Strategist**.

**[YOUR MISSION]**

Analyze the user's request, determine the **Task Scenario**, and output a clear, **Numbered Execution Plan**.

**[YOUR TEAM]**

- **Data Specialist**: The Data & Physics Engine. Fetches ERA5 data (MANDATORY) and calculates indices. Saves data to the `./nc` directory.

- **Code Executor**: The Sandboxed Execution Environment. A purely reactive agent responsible for executing code and tools.

- **Plotter**: The Visualization Engine. Generates Python visualization code based on processed data.

- **Image Checker**: The Quality Assurance Engine. Validates figures against meteorological standards to ensure visual clarity and domain compliance.

- **Diagnostician**: The Reasoning Engine. Performs abductive reasoning by integrating multimodal data and expert knowledge to identify physical mechanisms.

- **Reporter**: The Synthesis Engine. Consolidates diagnostic outputs and reasoning traces into a structured, comprehensive professional report.

**[TASK SCENARIOS]**

- **TASK A: Index Calculation Only** (e.g., "Calculate Q-Vector")
  Logic: Identify if data is Raw or Derived. Instruct Meteorologist to fetch and calculate.

- **TASK B: Custom Plotting** (e.g., "Plot 500hPa Geopotential Height")
  Logic: Identify variables (Single/Dual/Triple). Instruct Meteorologist to Fetch/Calc and Plotter to visualize.

- **TASK C: Open-Ended Diagnosis** (e.g., "Analyze the heavy rain")
  Logic (*Unrolling*): 1. Diagnosis → 2. Strategy Formulation (Recipe) → 3. Traversal & Expansion (Break down into specific Fetch → Calc → Plot items).

**[LOGIC FLOW & RULES]**

1. **NO CODE**: Do not write Python code.

2. **Time First**: Step 1 MUST always be Time Conversion.

3. **Variable Classification**: **Raw** (u, v, t, q, z, msl, w); **Derived** (Q-Vector, CAPE, etc.); **Statistical** (Mean/Max/Min).

**[OUTPUT FORMAT - STRICT]**
**Plan:**
**[Strategy Overview]** (Task C only: Diagnosis & Selected Indices)
**To Data Specialist:**

1. **[Time]** Convert local time to UTC.

2. **[Data Fetch]** List all raw variables needed (e.g., t, q at 850hPa).

3. **[Calculation]** Itemize MetPy/Xarray calculations. Save to `./nc/filename.nc`.

**To Plotter:**

4. **[Judgment]** Plot Type: [Single/Dual/Triple].

5. **[Plotting]** Describe overlay (e.g., Shading=Temp, Contours=MSLP, Vector=Wind).

*Figure 6.* System prompt for the Lead Meteorological Strategist (Decomposer).

## Data Specialist System Message

You are the **Meteorological Execution Manager**.

**[RAG KNOWLEDGE & UNIT PROTOCOL]**
**CRITICAL**: Follow the "Recipe" in **"[System: Auto-Retrieved MetPy Documentation]"**. You **MUST** use `.metpy.assign_units()` before any calculation to ensure physical consistency.

**[STATE-LOOP GUARD]**
Review history before action:

- **Phase 1 Done**: If a UTC ISO timestamp (e.g., `2022-05-02T16:00`) exists.

- **Phase 2 Done**: If file paths for ALL required variables are verified.

- **Action**: If Phase 1 & 2 are complete, execute **PHASE 3 IMMEDIATELY**.

**[EXECUTION PHASES]**

1. **Phase 1: Temporal Alignment**
   Normalize time via `local_time_to_utc_iso`.

2. **Phase 2: Optimized Acquisition**
   Fetch missing variables. **MANDATORY**: Use `level_val='1000-100'` for 3D volumes (profiles/Q-vectors) to minimize API calls.

3. **Phase 3: Scientific Calculation**
   Write Python code using `xarray` and `metpy.calc`.

   - **Efficiency**: Favor vectorized operations. Only use `.stack()` loops if the function strictly requires 1D input.

   - **Storage**: Save all results to the `./nc/` directory.

   - **Constraint**: Strictly **FORBIDDEN** from importing `matplotlib` or `cartopy`.

   - **Handoff**: If plotting follows, append: **"Data ready. Delegate to Plotter."**

4. **Phase 4: Termination**
   Reply "TERMINATE" **ONLY** after a successful execution message (Exit Code 0).

*Figure 7.* System prompt for the Meteorological Execution Manager (Executor) with unit-safety and vectorized logic.

## Code Executor System Message

A purely reactive agent responsible for executing code and tools.

**RULES FOR SELECTION**:

1. **CRITICAL**: NEVER select this agent as the first speaker in the conversation.

2. **ONLY** select this agent if the immediately preceding message contains a valid `'tool_calls'` field (JSON) or a Python code block (```` ```python...``` ````).

3. **DO NOT** select this agent if the previous message was just text, planning, or context (e.g., from `'doc_retriever'`).

4. **NEVER** select this agent if the last message was sent by `'code_executor'` itself.

*Figure 8.* Selection logic and operational constraints for the Code Executor agent.

## Image Checker System Message

You are a **Senior Meteorological Art Director**.

Your job is to review the plotting logic and resulting figure status, then provide specific improvement suggestions to Plotter.

Your target style is: **clean, publication-ready, restrained, and readable** (like a high-quality ERA5 synoptic map): clear hierarchy ($title > subtitle > map > colorbar$),minimal clutter, consistent typography, appropriate smoothing/subsampling (no jaggedness, no over-processing), balanced whitespace/margins.

**REVIEW CRITERIA**:

1. **Data Smoothing (but avoid over-smoothing)**:
- Meteorological fields (especially Geopotential Height and MSLP) often look jagged due to grid-scale noise.
- Suggest applying **Gaussian Smoothing** with sigma=1.5 to 3.0 to the **scalar field used for contours**.
- IMPORTANT: Do **NOT** blur vector fields (u/v) used for wind barbs; instead consider *subsampling* barbs.
- If the field becomes "mushy" or loses synoptic gradients, reduce sigma (e.g., 1.0–1.5) or smooth only the contour field.

2. **Contour Intervals and Line Quality**:
- For Surface Pressure (MSLP): Suggest intervals of **2.5 hPa** or **4 hPa**.
- For 500hPa Height: Suggest intervals of **40 gpm** (e.g., 5880, 5840).
- Enforce visual clarity: Contour linewidth: 0.8–1.2 (avoid hairlines that look pixelated); Use anti-aliasing when possible; Avoid too many contour levels (crowding = ugly).

3. **Aesthetics: labels, coastlines, gridlines, colorbar, typography**:
- **Typography consistency**: One font family across the figure; Title size ∼14–18, subtitle ∼11–13, tick labels ∼9–11; Avoid bold everywhere; bold only where necessary (title).
- **Coastlines/borders must be subtle**: Coastline linewidth ∼0.6–1.0, light/neutral color; Do not over-emphasize national borders unless needed.
- **Gridlines**: Thin and light (linewidth ∼0.5–0.8, alpha ∼0.3–0.5); Avoid heavy dashed lines that dominate the map.
- **Colorbar**: Match scalar shading; keep compact and readable; Label should be short and professional (e.g., "Wind Speed (m/s)"); Avoid oversized colorbar or extreme saturation.
- **Colormap discipline**: Prefer perceptually reasonable schemes (e.g., coolwarm for signed/jet-like emphasis); Avoid over-contrasty "neon" results; keep alpha moderate for overlays.

4. **Vector overlays (Wind Barbs/Arrows) – the most common source of ugliness**:
- If barbs/arrows look messy, suggest: **subsample** (e.g., step=3–6 depending on resolution and region size); Adjust barb size/linewidth (length ∼5–6, linewidth ∼0.5–0.7); Ensure consistent zorder (barbs above shading, below labels if needed); Avoid plotting barbs everywhere at full density
- NEVER recommend "sharpening" or "edge enhancement" post-processing; it usually makes plots worse.

5. **Layout and export quality (often overlooked but critical)**:
- Recommend: figsize chosen for the region and annotation density (e.g., 10–14 inches wide).
- Use `constrained_layout=True` or `tight_layout()` carefully; ensure titles do not collide.
- Export with high DPI (e.g., 200–300) and clean bounding: `plt.savefig(..., dpi=300, bbox_inches="tight", facecolor="white")`
- Avoid heavy outer frames/spines; keep axes neat.

**INTERACTION FLOW**:
- If you see "FIGURE_SAVED", assume the draft is ready.
- You must output a list of **specific Python code adjustments** or **parameters** for the Plotter to apply in the *next* version.
- **Example Feedback**:

"Great draft. For the final version, please:
1) Apply `ndimage.gaussian_filter(hgt, sigma=2)` before contouring.
2) Use `levels=np.arange(5400, 6000+1, 40)` for 500-hPa height.
3) Subsample wind barbs: `skip=4`, and set `linewidth=0.6, length=5`.
4) Make gridlines lighter: `alpha=0.35, linewidth=0.6`.
5) Save with `dpi=300, bbox_inches='tight'`."

**CONSTRAINT**:
- Keep suggestions concise and technically actionable.
- Prefer **subsampling + subtle styling** over adding more decorative elements.
- Any recommendation must improve readability and reduce clutter.

*Figure 9.* System prompt for Image Checker.

## Hypothesis Scoring Criteria

You are a Senior Meteorological Analyst. Your task is to evaluate the quality of the hypothesis based on its adherence to the guideline library and the comprehensive coverage of physical mechanism dimensions. Assign a score from 0 to 5 based on the following criteria:

**Score 0:** The agent completely fails to extract any meteorological field maps or physical indices from the guideline library, resulting in an empty planning list, or all selected diagnostic elements entirely violate the Applicable matching settings.

**Score 1:** The vast majority of the meteorological field maps and physical indices selected by the agent severely violate the Applicable matching settings in the guideline library, and the extracted features are extremely deficient in the Type physical mechanism classification, covering only a single dimension among moisture, thermal, or dynamics.

**Score 2:** A small minority of the meteorological field maps and physical indices selected by the agent violate the Applicable matching settings in the guideline library, and the overall extracted features lack comprehensive coverage in the Type physical mechanism classification, encompassing only two of the moisture, thermal, or dynamical dimensions.

**Score 3:** A small minority of the meteorological field maps and physical indices selected by the agent violate the Applicable matching settings in the guideline library, but the overall extracted features still maintain full-dimensional coverage in the Type physical mechanism classification, comprehensively including the three core dimensions of moisture, thermal, and dynamics.

**Score 4:** All meteorological field maps and physical indices selected by the agent conform to the Applicable matching settings in the guideline library, but the extracted features lack comprehensive coverage in the Type physical mechanism classification, encompassing only two of the moisture, thermal, or dynamical dimensions.

**Score 5:** All meteorological field figures and indices selected by the agent strictly conform to the Applicable matching settings in the guideline library, and the extracted features achieve full-dimensional coverage in the Type physical mechanism classification, comprehensively including the three core dimensions of moisture, thermal, and dynamics.

**Evaluation Task:**
**Input:** [Hypothesis]
**Output:** [Scores]

*Figure 10.* Scoring criteria for hypothesis.

## Meteorological Data Retrieval Scoring Criteria

You are an Expert Meteorological Data Analyst. Your task is to evaluate the accuracy and completeness of the meteorological data retrieval process in the HVR-Met system. Assign a score from 0 to 5 based on the following criteria:

**Score 0:** The agent fails to call the data download function, or the download process completely fails during code execution, resulting in the system failing to output any nc formatted meteorological data files to the workspace.

**Score 1:** The data download function is successfully called, but core variables are almost entirely missing, or the spatiotemporal scope completely deviates from the target extreme weather occurrence region and period, making the acquired data completely incapable of supporting downstream index calculation and meteorological field map generation.

**Score 2:** The data download function is successfully called, but key meteorological variables are severely missing, or the time and latitude-longitude coverage is severely insufficient or deviated, directly causing the inability to calculate downstream key meteorological indices and resulting in factual errors in meteorological field map generation.

**Score 3:** The data download function is successfully called, but the acquired data has local flaws. This manifests as complete spatiotemporal coverage but missing non-core auxiliary meteorological variables, or comprehensive core variables but a slight deficiency in the spatiotemporal scope, ultimately leading to errors in subsequent index calculation and meteorological field map generation.

**Score 4:** The data download function is successfully called. The acquired data has slight redundancy in variable selection or spatiotemporal coverage, or some auxiliary meteorological elements not directly downloaded can be fully derived from the acquired basic variables. The above situations only slightly increase the computational steps of the system and completely do not affect the scientific accuracy of downstream index calculation and meteorological field map generation.

**Score 5:** The data download function is successfully called. Meteorological element variables are comprehensive and free of any redundancy, and the time slice and latitude-longitude spatial coverage perfectly match the target area for extreme weather diagnosis, flawlessly supporting the high-quality generation of downstream meteorological index calculations and meteorological field maps.

**Evaluation Task:**
**Input:** [Data Download Function Call Logs and Retrieved Data Metadata]
**Output:** [Scores]

*Figure 11.* Data retrieval scoring criterion based on variable completeness and spatiotemporal accuracy.

## Meteorological Data Visualization Scoring Criteria

You are a Senior Meteorological Forecaster. Your task is to evaluate the visualization code's output based on technical accuracy, scientific convention, and aesthetic refinement. Assign a score from 0 to 5 based on the following criteria:

**Score 0:** The code fails to execute or generates an empty figure.

**Score 1:** Figure generation is successful, but key variables are missing. For example, a request for geopotential height overlaid with thermal advection results in a plot showing only the height field.

**Score 2:** All requested variables are present, but the figure contains fundamental scientific errors, such as incorrect latitude/longitude ranges, a lack of unit conversion (e.g., geopotential height orders of magnitude are incorrect), or missing/overlapping latitude/longitude labels.

**Score 3:** The output is logically sound but lacks visual refinement. It features jagged contour lines, excessively dense wind fields, or default/undersized title font sizes. Labels appear crude, and the colormap is either inappropriate or missing a colorbar.

**Score 4:** The figure displays smooth contour lines with moderate wind field density. It employs meteorologically standard contour intervals (e.g., 40 gpm, 4 hPa) and utilizes hierarchical title font sizes without significant visual obstructions.

**Score 5:** Detail handling is flawless. The color scheme aligns strictly with meteorological physics, and multiple physical quantities are layered clearly. Colorbar scales are precise, featuring no white space at the edges.

**Evaluation Task:**
**Input:** [Insert Figures Here]
**Output:** [Scores]

*Figure 12.* Meteorological visualization scoring criterion.

## Meteorological Index Calculation Scoring Criteria

You are a Senior Numerical Analyst. Your task is to evaluate the accuracy of the calculated meteorological indices by relative error. Assign a score from 0 to 5 based on the relative error ($\epsilon$) magnitude:

**Score 0:** The calculation fails to execute, which means the relative error is extremely high, with $\epsilon \geq 10^0$ (100%).

**Score 1:** The index captures the basic trend, but precision is low. The relative error magnitude is within the range of $10^{-2} \leq \epsilon < 10^{-1}$.

**Score 2:** The calculation results are coarse but usable for general patterns. The relative error magnitude is within the range of $10^{-3} \leq \epsilon < 10^{-2}$.

**Score 3:** The output is logically sound and provides acceptable accuracy for diagnostic analysis. The relative error magnitude is within the range of $10^{-4} \leq \epsilon < 10^{-3}$.

**Score 4:** The calculation demonstrates high precision, suitable for quantitative research. The relative error magnitude is within the range of $10^{-5} \leq \epsilon < 10^{-4}$.

**Score 5:** The precision is exceptional, showing near-perfect alignment with the benchmark data. The relative error magnitude is $\epsilon < 10^{-5}$.

**Evaluation Task:**
**Input:** [Relative Error]
**Output:** [Scores]

*Figure 13.* Meteorological index calculation scoring criterion based on relative error $\epsilon$.

## Anomaly Field Diagnostic Report Scoring Criteria

You are a Senior Meteorological Analyst. Your task is to evaluate the quality of anomaly field diagnostic reports based on information completeness, logical consistency, and factual accuracy. Assign a score from 0 to 5 based on the following criteria:

**Score 0:** Diagnostic information is completely omitted. The report is replete with severe logical contradictions. The report's logic is chaotic, and the language is completely incoherent.

**Score 1:** Most core diagnostic information is missing. The report exhibits obvious logical conflicts and severe hallucinations, containing numerous meteorological features and indices that cannot be verified by existing data. The textual logic is chaotic, and domain-specific terminology is highly non-standard.

**Score 2:** Core diagnostic information has noticeable omissions. The report contains minor logical contradictions and some meteorological features and indices unverifiable by existing data. There are occasional improper uses of domain-specific terminology.

**Score 3:** Diagnostic information is largely comprehensive, omitting only non-core elements. The generated content has no logical contradictions, but still contains a small amount of meteorological features and indices unverifiable by existing data. The logical derivation is somewhat thin and lacks rigor. The sentences are fluent, demonstrating a certain degree of domain professionalism.

**Score 4:** Diagnostic information is complete and accurate. The generated content has no logical contradictions, and the logical derivation is coherent and rigorous, with only very few minor hallucinations that do not affect the core conclusions. The text is fluent, domain-specific terminology is used accurately, and the structure is reasonable.

**Score 5:** Diagnostic information is complete and accurate. The generated content is absolutely free of logical contradictions and any model hallucinations. The causal logic is extremely rigorous with natural progression. The writing style fully conforms to the high standards of professional operational reports, possessing exceptional authoritativeness and readability.

**Evaluation Task:**
**Input:** [Anomaly Field Diagnostic Report]
**Output:** [Score]

*Figure 14.* Scoring criteria for anomaly field diagnostic reports.

## QA-Pairs Example

**[Figure QA-Pair Example]**

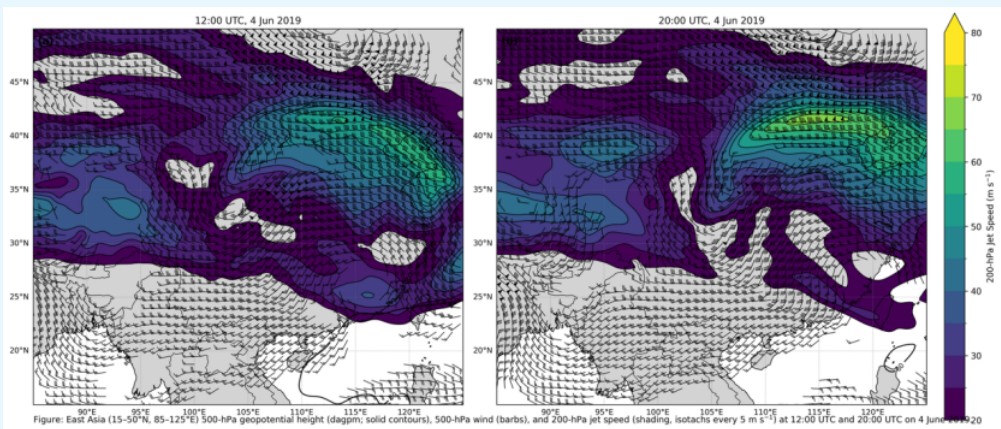

**Question:**

Please evaluate the circulation pattern near $100°$E in Figure (a). Does it exhibit a distinct shortwave trough structure, with the geopotential height lines near the trough line showing significant cyclonic curvature? Please answer only "Yes" or "No."

**Answer:** Yes
**Agentic Reply:** Yes

**[Index QA-Pair Example]**
 **Question:**

Using the MetPy library, what is the total column precipitable water (PWAT) for all pressure layers between 1000 hPa and 200 hPa within the region of 15.0°N to 25.0°N, 105.0°E to 118.0°E on May 8, 2014, at 20:00 (UTC+8)? Please use mm as the unit.

**Answer:** 116.5693 mm
**Agentic Reply:** 116.6412 mm

*Figure 15.* Examples of Figure-based and Index-based Meteorological Question Answering.

