# OpenReview forum: "HVR-Met: A Hypothesis-Verification-Replanning Agentic System for Extreme Weather Diagnosis"
_ICML.cc/2026/Conference — ICML 2026 regular_

### Official Review · Reviewer_TViE · 2026-03-09

**Soundness:** 3
**Presentation:** 2
**Significance:** 2
**Originality:** 3
**Overall Recommendation:** 4
**Confidence:** 4

**Summary:**

This study presents HVR-Met, a multi-agent system designed for professional-level extreme weather detection. The main goal is to turn meteorological expert procedures into a Hypothesis-Verification-Replanning loop, with agents for planning, getting data, running code, charting, checking images, diagnosing, and making reports. The system is further backed up by a library of meteorological guidelines and tool-oriented knowledge bases that are built semi-automatically. The authors also present a benchmark that includes end-to-end diagnosis, calculating meteorological indices, and making figures.

**Compliance With Llm Reviewing Policy:**

Affirmed.

**Final Justification:**

The authors have fully addressed my concerns. I am considering raising my score to 3. My remaining reservations are mainly due to the limited information regarding evaluation across different LLMs, and the application scenario of extreme climate prediction may still need further confirmation as to whether it warrants acceptance.

**Key Questions For Authors:**

Major:

1. The report says that the 100 extreme weather occurrences that happened from start to finish are not related to the 584 papers that were used to create the knowledge base. But a lot of these previous catastrophic weather events may have still shown up in the pretraining material of the closed-source LLMs that were used to train them. Consequently, it remains uncertain whether robust performance signifies the HVR reasoning mechanism or the retention of previously encountered meteorological analyses and reports.

To further separate the effect of HVR-Met, the authors should control for the dates when the basis model was released and trained, and, if possible, test it on occurrences of extreme weather that happened only after those dates. This kind of time split would make the claims of reliability much stronger.

2. There is insufficient support for the claim of high correlation between expert scores and LLM evaluation. In Section 4, the authors state: “The high correlation observed between the expert manual scores and the LLM evaluations validates the operational accuracy of our automated scoring workflow.” However, the paper does not specify:

- which correlation metric was used (e.g., Pearson, Spearman, Kendall);
- what the actual correlation value is;
- how agreement varies across scoring dimensions.

In addition, only 20 out of 100 end-to-end cases were manually scored. For an ICML paper making evaluation-validity claims, this sample size feels too limited, especially without detailed error analysis. The paper should report the exact statistics and ideally expand the human validation beyond 20 samples, or at least justify the sampling design and provide disagreement case studies.

3. The physical causes of extreme weather are highly region- and season-dependent. For example, even within the same label such as “rainstorm,” monsoon-driven rainstorms during flood season and summer cold-vortex rainstorms can have very different physical mechanisms and diagnostic indices.

It is therefore important to clarify whether the Guideline Library includes region and season as explicit conditional branches, and whether the benchmark/test set is similarly stratified. Without this, it is hard to assess whether the framework truly captures expert diagnostic diversity or mainly learns generic templates.

4. The eplanning trigger and logical consistency are not sufficiently operationalized. The paper emphasizes that the system detects discrepancies and triggers replanning when evidence conflicts with the initial hypothesis. However, the actual trigger conditions remain underspecified. For example, if the VLM incorrectly identifies a trough as a ridge, or misses a vortex structure, does the system automatically enter replanning? How is “logical consistency” formally judged in the framework?
Since the HVR loop is the central technical contribution, the paper should make the replanning policy more explicit, including failure cases and the robustness of the pipeline to upstream VLM errors.

Minor Comments
1. Please correct the typo “Automic-level” to “Atomic-level.”
2. Please double-check the subsection titles: both 5.4 and 5.5 are labeled “Ablation Study.”
3. I strongly encourage the authors to include in the appendix a complete high-quality example of an agent-generated diagnostic report, ideally side-by-side with a human forecaster’s report, to better illustrate qualitative strengths and remaining gaps.

**Limitations:**

The paper has several important limitations include limited evaluation scale for a safety-critical application, insufficient control for potential memorization from pretrained models, lack of analysis on regional/seasonal variability in weather mechanisms, and insufficiently specified replanning and discrepancy-detection criteria in the HVR loop.

**Strengths And Weaknesses:**

Strengths

1. The study describes a domain-specific event-driven pipeline that includes generating hypotheses, verifying them with several methods, finding discrepancies, and using feedback to plan again. Adding multimodal building and assessment logic is a big step forward from single-model techniques, which typically have trouble doing reliable meteorological reasoning.

2. The guideline library is built through extraction, aggregation/consolidation, semantic completion, and expert verification. This is a sensible knowledge engineering design that aligns with how operational meteorological expertise is typically formalized.

3. The dual-track validation idea for weather figures is a notable contribution. In particular, using VLM-based QA on physically meaningful visual patterns (e.g., vortices, troughs, shear lines) is more appropriate than naive pixel similarity, and provides a plausible way to evaluate the scientific validity of generated plots.

Weaknesses

1. Experimental evidence is still relatively weak for such a high-stakes application. For a task like extreme weather diagnosis, where both false positives and false negatives can be costly, the current experiments mainly demonstrate the potential value of the multi-agent framework rather than its current reliability in realistic deployment settings.

---

> ### Author Rebuttal · Authors · 2026-03-31
>
> We sincerely thank the reviewer for the insightful comments. We have carefully addressed each of the concerns raised above, and we respectfully hope that the reviewer will reconsider the evaluation in light of these clarifications and provide a positive assessment.
>
> Q1:
> Among the 100 test cases, 22 are drawn from publicly available literature, all published after the model's release date. The remaining 78 are sourced from internal materials provided by domain experts and are not publicly accessible. We are therefore highly confident that our framework demonstrates genuine, robust performance. All test cases will be made publicly available.
>
> Q2:
> We report Pearson correlation coefficients across all five scoring dimensions, computed on an expanded set of N=30 human-annotated cases (up from 20). Detailed per-dimension r and p-values are provided at the anonymous link: https://tinyurl.com/4h65spjc. All dimensions achieve r $≥$ 0.81 (p $\ll$ 0.01), confirming strong positive correlation between expert and LLM-based scoring. Index stage achieves the highest correlation due to its objectively verifiable ground truth. Data stage yields near-perfect agreement, as most models performed correctly on this dimension, resulting in minimal scoring variance.
>
> Q3:
> To verify that the framework captures expert diversity rather than applying generic templates, we encode each case as a 50-dimensional binary vector (20 meteorological fields and 30 diagnostic indices; 1 indicates selected, 0 indicates not selected). We analyze all the cases across three precipitation subtypes(K=3): convective precipitation, cyclonic precipitation, and orographic precipitation.
>
> Intra-class diversity: All vectors within each subtype are mutually distinct, confirming that the framework generates case-specific selections rather than applying a fixed template.
>
> Inter-class separability: We employ PERMANOVA [1] to assess separability via the pseudo-F statistic, where $SS_B$ and $SS_W$ denote between-class and within-class dispersion, respectively:
>
> $$F^* = \frac{SS_B/(K-1)}{SS_W/(n-K)} = \frac{1.9778/2}{3.53370/17} = 4.7531$$
>
> The permutation test yields p=0.0001 ($<$0.05), indicating that inter-class differences are significantly greater than intra-class differences. These results confirm that the framework faithfully reflects the diversified reasoning of domain experts and adapts its analysis flexibly across distinct rainstorm subtypes.
>
> [1] Anderson, M.J., et al. "A new method for non-parametric multivariate analysis of variance." Austral Ecology 26.1 (2001): 32–46.
>
> Q4:
> Regarding the replanning trigger mechanism, we operationalize it at two levels.
>
> At the index level, we adopt Standardized Anomaly (SA) as the quantitative criterion, where X is the current computed value, and $\mu$ and $\sigma$ are the climatological mean and standard deviation over the 1990–2019 period, respectively. An absolute SA value exceeding 2 is classified as a significant anomaly.
>
> At the figure level, the Diagnostician performs targeted feature inspection on generated figures based on verification objectives preset in the Guideline Library. The inspection content is constrained by domain knowledge rather than open-ended VLM reasoning. Replanning is triggered when any index or figure fails to exhibit the expected anomaly. Additionally, the multi-agent system enforces a maximum interaction limit of 50 rounds to prevent infinite loops.
>
> Regarding the definition of "logical consistency": given the current hypothesis, the system expects a set of indices and figures to exhibit significant anomalies. After verification, if all items meet the expected anomalies, logical consistency is confirmed and the hypothesis is accepted; if any item fails to exhibit the expected anomaly, logical inconsistency is determined and the HVR loop triggers replanning.
>
> Regarding robustness to VLM errors, we employ GPT-4V as the model for diagnosing meteorological imagery. Related work [2] has demonstrated that GPT-4V can identify meteorological structures (troughs, ridges, etc.) with high accuracy, showing strong agreement with human-issued forecasts. We acknowledge that VLM misidentification remains a potential risk, and we plan to introduce cross-validation with multiple VLMs in future work to further enhance robustness.
>
> [2] Lawson, John R., et al. "Pixels and predictions: potential of GPT-4V in meteorological imagery analysis and forecast communication." Artificial Intelligence for the Earth Systems 4.1 (2025).
>
> Furthermore, we appreciate the reviewer's careful attention to detail. We accept all minor comments and have corrected the identified typos accordingly. In addition, we will provide a complete high-quality example of an agent-generated diagnostic report side-by-side with an expert forecaster's report in the appendix of the revised manuscript.

---

> > ### Author Rebuttal · Reviewer_TViE · 2026-04-03
> >
> > The authors have fully addressed my concerns. I am considering raising my score to 3. My remaining reservations are mainly due to the limited information regarding evaluation across different LLMs, and the application scenario of extreme climate prediction may still need further confirmation as to whether it warrants acceptance.

---

> > > ### Author Response · Authors · 2026-04-06
> > >
> > > We sincerely thank you for your positive feedback and are delighted that our previous response has fully addressed your concerns. To resolve your remaining reservations, we provide a comprehensive evaluation across different LLMs below, and further clarify the critical real-world value and broad research implications of HVR-Met in the extreme weather diagnosis scenario.
> > >
> > > ### **Q1:**
> > >
> > > **For extreme weather diagnostic task across different LLMs (Table 1):**
> > >
> > > GPT-5 and Gemini-3-Pro-Preview-Thinking achieved high scores of 4.70 and 4.63, respectively. The HVR-loop iterations were completed within just 3 rounds.This demonstrates that these models possess extensive prior knowledge in meteorology and high-level strategic planning capabilities. They can accurately anchor the correct physical causal pathways and achieve rapid convergence of complex reasoning processes with minimal trial-and-error costs.
> > >
> > > During the index calculation and figure generation phase, the scores of Qwen3-Coder and Deepseek-R1 were significantly lower. Execution logs reveal that when encountering dimension or unit errors while calling specialized meteorological libraries (e.g., MetPy, Cartopy), these two models lacked the error-correction resilience and strict tool adherence required for complex APIs. After only one or two rounds of trial and error, they gave up on fixing the parameters and fell back to a generic code comfort zone, ultimately leading to computational failure and a comprehensive degradation of output quality.
> > >
> > > **For sub-task analysis across different LLMs (Figure 4):**
> > >
> > > As Figure 4(a) shows, reasoning-enhanced models (Gemini-thinking: 46.43%, DeepSeek-R1: 42.31%) outperform GPT-5 (31.25%) on hard-level indices requiring multi-variable retrieval and multi-step derivations. They achieve this by accurately decomposing complex formulas and iteratively calling numerical libraries (e.g., MetPy), which avoids internal approximations and eliminates error accumulation in long chains. In addition, code-specialized models such as Qwen3-Coder (26.32%) significantly lag behind general and reasoning models in physics-based calculations. Log analysis reveals that their primary failures stem from dimensional handling errors during complex unit conversions. This highlights that despite strong coding skills, their inadequate understanding of physical concepts remains a critical bottleneck in index computing.
> > >
> > > As Figure 4(b) shows, plotting accuracy declines sharply with difficulty (e.g., GPT-5 drops from 96.31% to 54.66% on Hard tasks). In these tasks requiring multi-layer overlays, most errors are not numerical, but visual occlusions. Improper management of multi-layer rendering order (z-order) obscures critical information, rendering the plots ineffective for meteorological analysis.
> > >
> > > ### **Q2:**
> > >
> > > We provide the following explanations regarding the importance of HVR-Met’s application scenarios:
> > > 1. **Accurate diagnosis of extreme weather is the fundamental basis for ensuring public safety.** Currently, extreme weather events are becoming increasingly frequent, posing severe threats to lives, property, and agricultural production. Therefore, improving the accuracy of extreme weather prediction has become a matter of urgency.
> > > 2. **Aligning with operational diagnostic workflows.** Unlike end-to-end models that fail to replicate expert reasoning, HVR-Met perfectly matches practical requirements. It delivers precise extreme weather diagnoses supported by visualized multi-modal evidence and meteorologically sound causal reasoning.
> > > 3. **Empowering human forecasters with Expert Knowledge for High-Level Diagnostics.** Extreme weather diagnosis is highly dependent on long-term personal experience. The core value of HVR-Met lies in its ability to effectively absorb and formalize diagnostic logic and domain knowledge from professional literature and senior forecasters. Rather than relying on rigid templates, the system utilizes the HVR-loop to ensure that every routine meteorological diagnosis flexibly and accurately reaches the competency level of a junior forecaster.
> > > 4. **Continuous Evolution of the Expert Knowledge Base.** The long-term value of HVR-Met extends beyond its current diagnostic capabilities. We anticipate that during real-world deployment, every successful reasoning path and diagnostic report, once verified by human experts, can be integrated back into the system as new "expert knowledge". This enables the system to continuously self-iterate and adapt to the highly complex and dynamic nature of extreme weather scenarios.
> > >
> > > Once again, thank you for your insightful feedback and willingness to reconsider the evaluation. We hope the comprehensive response above fully addresses all your remaining questions. Based on our recent reply, we respectfully request that you re-evaluate our work and consider raising your final score.

---

### Official Review · Reviewer_YQgX · 2026-03-12

**Soundness:** 3
**Presentation:** 3
**Significance:** 3
**Originality:** 3
**Overall Recommendation:** 4
**Confidence:** 4

**Summary:**

This paper aims to develop a multi-agent system for extreme weather diagnostics. The main challenge is that AI agentic systems are hindered by a lack of adequate expert knowledge and high-grade reasoning loops and even validation and evaluation systems under extreme conditions. They propose a multi-agent meteorological diagnostic system that integrates expert knowledge and utilizes a closed loop mechanisms that enables iterative reasoning.

**Compliance With Llm Reviewing Policy:**

Affirmed.

**Final Justification:**

I maintain my Weak Accept recommendation. The paper addresses an important and timely problem, developing a multi-agent framework for extreme weather diagnostics, and proposes an interesting closed-loop Hypothesis-Verification-Replanning (HVR) mechanism inspired by human forecasters. The manuscript is generally well structured, clearly written, and the system-level contribution is meaningful. The integration of domain knowledge, role-specialized agents, and iterative reasoning is a promising direction and could be useful for future research in agentic scientific workflows.

However, some of my main concerns remain only partially addressed after the rebuttal. In particular, the theoretical grounding of the proposed framework is still limited. While the authors clarified that termination rather than convergence is the appropriate formal property and provided a finite hypothesis-space argument, this does not fully establish the reliability or correctness of the reasoning mechanism itself. Thus, the soundness of the framework remains moderately supported but not rigorously justified.

The authors did respond constructively to other concerns. They acknowledged the need to revise the Introduction to better distribute citations and committed to adding a discussion of limitations and potential societal impacts. These responses are helpful and improve confidence in the final version, but they are largely commitments for revision rather than issues fully resolved in the current manuscript.

Balancing strengths and weaknesses:

* Soundness: Moderate; system design is reasonable but theoretical grounding remains limited.
* Originality: Good; the HVR-based multi-agent diagnostic loop is a novel and interesting formulation.
* Significance: Good; the approach targets an important real-world application and could influence future work.
* Clarity: Good; overall presentation is clear, with minor issues identified.

Overall, the rebuttal partially addressed my concerns but did not substantially change my evaluation. I therefore maintain my Weak Accept recommendation, viewing this as a technically interesting and relevant contribution whose impact would be strengthened with more rigorous theoretical justification and expanded discussion of limitations.

**Key Questions For Authors:**

1.Please clarify the foundational principles of the proposed multi-agent framework.
2.How exactly is the multi-agent component structured and executed, and
3.are there supporting mathematical proofs to validate its theoretical soundness?

**Limitations:**

This paper would be greatly improved by clearly discussing the limitations of the proposed HVR-Met framework. Please include an analysis of the framework's constraints and any potential negative societal impacts, as this transparency is essential for future researchers building upon this work.

**Strengths And Weaknesses:**

This manuscript is clearly written, well-structured, and fairly easy to follow. Additionally, the authors are addressing a highly interesting and relevant problem. Replicating the analytical discipline of human forecasters through a continuous cycle of hypothesis generation, verification, and strategic replanning is an interesting and compelling approach.

However, there is an area for improvement regarding how the prior literature is presented in the Introduction. Currently, the first two paragraphs introduce the topic but lack any citations, while all the references are heavily condensed into the third paragraph. This structure makes it difficult to connect the specific background claims to their corresponding sources.

I suggest revising the Introduction to distribute the citations more evenly. Please ensure that the background information and statements made in the first two paragraphs are immediately supported by the appropriate references, rather than grouping all citations together later in the text.

---

> ### Author Rebuttal · Authors · 2026-03-31
>
> Q1: The foundational principles of HVR-Met can be summarized in four aspects:
>
> First, it emulates the cognitive workflow of human experts in the professional "Weather Consultation" process, decomposing the procedure by which senior forecasters identify anomalous signals and perform attribution reasoning under strict time constraints into automatable, role-specialized sub-tasks.
>
> Second, it achieves deep integration of domain-specific expertise. The system semi-automatically extracts key diagnostic elements from 584 extreme weather papers, rigorously validated by five senior forecasters, to construct a Guideline Library encompassing four dimensions: knowledge category, modality, applicable weather types, and standardized descriptions. This provides the agents with professional-grade prior knowledge.
>
> Third, it introduces the "Hypothesis--Verification--Replanning" (HVR) closed-loop mechanism. The system formulates testable physical hypotheses grounded in anomalous signals, generates multi-modal evidence through index computation and synoptic figure rendering, and evaluates the alignment between evidence and hypotheses. If a discrepancy is detected, the system updates its short-term memory, queries the Guideline Library for alternative causal mechanisms, and triggers a new verification cycle, thereby achieving self-correction of the diagnostic pathway.
>
> Fourth, it employs structured role specialization, decomposing the end-to-end diagnostic workflow into sub-tasks such as data acquisition, computation, visualization, quality assurance, and report generation, each handled by a dedicated agent augmented with domain knowledge through RAG modules.
>
> Q2: HVR-Met orchestrates seven specialized agents and its execution follows the four-phase HVR closed-loop:
>
> (1) Event-Driven Hypothesis Generation: the Data Specialist invokes the anomaly detection tool to identify statistical extremes and classify the event type, then retrieves corresponding diagnostic templates from the Guideline Library to formulate an initial physical hypothesis and an executable task chain.
>
> (2) Multi-Modal Verification Execution: the Data Specialist calculates quantitative indices while the Plotter renders synoptic figures, transforming abstract hypotheses into tangible multi-modal evidence.
>
> (3) Discrepancy Detection \& Reasoning: the Diagnostician evaluates the generated evidence against the initial hypothesis through visual-physical alignment.
>
> (4) Feedback-Driven Replanning: if the evidence corroborates the hypothesis, the findings are synthesized into a final diagnostic report by the Reporter; if the evidence fails to exhibit the expected anomalies or contradicts the hypothesis, the system formally rejects the current hypothesis, updates its short-term memory to record the negative result, queries the Guideline Library for alternative causal mechanisms, and instigates a new verification cycle until a physically consistent explanation is established.
>
> Q3: Our framework possesses theoretical soundness. Related mathematical research has demonstrated that multi-agent systems achieve optimal performance when aligned with a global objective [1].
>
> [1] Riedl, C., et al. "Emergent coordination in multi-agent language models." International Conference on Learning Representations (2026).

---

> > ### Author Rebuttal · Reviewer_YQgX · 2026-03-31
> >
> > 1. The response to Q3 is overly general. Citing external literature on emergent coordination does not constitute a proof of the theoretical soundness of the specific HVR-Met architecture. I was looking for a more rigorous justification of the convergence or reliability of your specific closed-loop mechanism.
> > 2. The authors did not acknowledge or commit to the requested revision of the Introduction. Distributing citations to support background claims is essential for the scholarly integrity of the paper.
> > 3. The request for a dedicated discussion on limitations and potential negative societal impacts (e.g., risks of automated weather diagnostics in emergency response) remains unaddressed.

---

> > > ### Author Response · Authors · 2026-04-01
> > >
> > > Thanks for your comment. We trust that these detailed clarifications have fully addressed your remaining concerns. We would be grateful if you would consider upgrading your recommendation for our paper.
> > > # Q1
> > > Sorry for the missing information. Due to character limit, our initial response omitted these details.
> > >
> > > We respectfully clarify that in the context of exploratory multi-agent systems, the relevant formal property to demonstrate is termination rather than convergence. Convergence is typically defined for iterative optimization or learning processes with continuous parameter updates, whereas our HVR loop operates as a physically constrained finite-state search over a bounded hypothesis space. Specifically, each candidate diagnostic factor is drawn exclusively from the Guideline Library, and each exploration step must retrieve factors matching the identified extreme weather type while ensuring coverage across moisture, thermodynamic, and dynamic dimensions. We provide a formal discussion of termination and reliability below.
> > > ## Definitions of Termination
> > > ### Definition 1 (Hypothesis Space)
> > > Let $\mathcal{H}$ denote the set of all diagnostic factors from the Guideline Library. This space is strictly finite:
> > > $$|\mathcal{H}| = N < \infty$$
> > > where $N$ is the number of all diagnostic factors in the Guideline Library.
> > > ### Definition 2 (System Objective and State)
> > > The goal of the HVR-Met framework is to obtain $B = 5$ anomalous factors. The state of the system at the $t$-th iteration is defined as:
> > >
> > > $B^{(t)}$: the set of anomalous factors verified up to the $t$-th round, where $0 \le |B^{(t)}| \le B$.
> > >
> > > $\mathcal{U}^{(t)} \subseteq \mathcal{H}$: the unexplored set, recording factors in the library that have not yet been sampled.
> > >
> > > Initially, $|B^{(0)}| = 0$ and $|\mathcal{U}^{(0)}| = N$.
> > > ## State Transition Process
> > > In the $t$-th iteration, if $|B^{(t)}| < B$, the system needs to sample a set of factors $\mathcal{N}^{(t)}$ from the unexplored set $\mathcal{U}^{(t)}$ to fill the gap:
> > > $$|\mathcal{N}^{(t)}| = \min\bigl(B - |B^{(t)}|, |\mathcal{U}^{(t)}|\bigr)$$
> > > According to the strictly sampling without replacement mechanism, regardless of whether these factors are ultimately verified, they are removed from the available space. Therefore, the unexplored set for the next round is updated as:
> > > $$\mathcal{U}^{(t+1)} = \mathcal{U}^{(t)} \setminus \mathcal{N}^{(t)}$$
> > > ## Description of Termination States
> > > The system will eventually achieve termination in one of the following three states:
> > >
> > > Success: When $|B^{(t)}| = B$, the system has collected all $5$ anomalous factors and successfully outputs the result.
> > >
> > > Exhaustion: When $|\mathcal{U}^{(t)}| < B - |B^{(t)}|$ and $|B^{(t)}| < B$, the unexplored space is depleted, and the system cannot collect $5$ anomalous factors, triggering an abnormal exit.
> > >
> > > Maximum Interaction Turns: The number of iterations $t$ reaches a preset threshold $T_{\max} = 50$, forcing termination to ensure engineering safety.
> > >
> > > In terms of reliability, we performed a rigorous statistical analysis. All cases successfully produced a final report, and $95$% identified five anomalous factors, sufficiently demonstrating the system's reliability.
> > > # Q2
> > > We sincerely thank the reviewer for valuable suggestions. We naturally adopt all feedback aimed at improving the writing quality of our manuscript, and we apologize for not explicitly communicating this in our previous response. We hereby assure you that the entire text will be carefully revised in the final version, strictly in accordance with your recommendations. We will expand our citations by adding studies on the timeliness challenges of nowcasting at the end of the first paragraph, and integrating relevant research on extreme weather diagnostic workflows within the second paragraph.
> > > # Q3
> > > We appreciate this valuable feedback and discuss HVR-Met's limitations and potential societal risks below.
> > >
> > > In practical weather diagnosis, forecasters typically synthesize data from multiple sources, such as the CMA’s GRAPES model, NOAA’s GFS model, and the ECMWF model. However, numerical weather prediction (NWP) models contain inherent errors. Even for the same weather event, different types of numerical models may produce starkly contrasting forecast outcomes. In operational practice, experienced forecasters often mitigate the limitations of single-source data by integrating their expertise with local observations.
> > >
> > > HVR-Met inherently depends on input data quality. Since our cases are all historical events, we used reanalysis data, which has largely eliminated the errors present in raw forecast data. However, in real-time operational applications, the inherent errors of raw forecast data will become a significant limitation. These inherent errors may lead to the omission or misjudgment of key weather features, resulting in negative social impacts such as interference with emergency dispatching. This limitation could potentially be mitigated through human-machine collaboration.

---

### Official Review · Reviewer_W4nN · 2026-03-13

**Soundness:** 2
**Presentation:** 2
**Significance:** 3
**Originality:** 2
**Overall Recommendation:** 2
**Confidence:** 4

**Summary:**

The paper introduces HVR-Met, a multi-agent system designed to automate extreme weather diagnosis. The system employs seven specialized agents (e.g., Decomposer, Data Specialist, Plotter) utilizing an AG2 framework , supported by a RAG-based knowledge library compiled from 584 meteorological papers.

**Compliance With Llm Reviewing Policy:**

Affirmed.

**Key Questions For Authors:**

See Strengths And Weaknesses.

**Limitations:**

While the system is a neat application of LLM agents to meteorology, the paper reads more like a systems engineering report or an application paper than a machine learning research paper. The lack of fundamental ML novelty, combined with reliance on existing APIs and frameworks, makes it a poor fit for the scientific standards of ICML.

**Strengths And Weaknesses:**

1. In my humble opinion, the core methodological contribution: the "Hypothesis-Verification-Replanning" (HVR) loop does not present a novel machine learning advancement. The mechanism relies heavily on existing multi-agent paradigms (using the AG2 framework) and standard LLM self-reflection and tool-use techniques. Mapping generic planning, execution, and verification steps to meteorological terminology (e.g., calling the planner a "Decomposer" or the tool-user a "Data Specialist") constitutes an application of existing AI concepts to a new domain, rather than a fundamental contribution to machine learning literature.

2. The framework primarily acts as a wrapper around state-of-the-art foundation models. The results demonstrate that the system's performance is heavily bound by the underlying capabilities of models like GPT-5 and Gemini-3-Pro-Preview-Thinking. The paper does not introduce any new model architecture, specialized fine-tuning technique, or novel mechanism for handling spatiotemporal data representations beyond generating Python code for existing libraries.

3. The evaluation of the "meteorological semantic integrity" for the figure generation sub-task heavily relies on Vision-Language Models (VLMs) answering binary questions about the generated plots. Because the system's internal "Image Checker" and "Diagnostician" agents also rely on similar Vision-Language capabilities to verify and replan, the evaluation metric risks being circular. An LLM generating code to satisfy a VLM judge does not robustly prove scientific validity without a much larger, purely human-expert evaluation beyond the 20% alignment check mentioned.

4. The system's success appears largely predicated on extensive, heavily engineered system prompts (as seen in the Appendix, e.g., strict constraints, formatting rules, and tool descriptions). While prompt engineering is necessary for agentic systems, a paper relying on rigid prompt templates as its primary mechanism for "expert knowledge integration" limits its generalizability and theoretical contribution.

---

> ### Author Rebuttal · Authors · 2026-03-31
>
> We sincerely thank the reviewer for the thoughtful comments. We would like to respectfully note that this paper is submitted to the Application-Driven Machine Learning track of ICML, which focuses on practical, end-user-driven problems in domains such as climate and physical sciences. Our work is motivated by the real-world needs of extreme weather diagnosis in climatology, and contributes domain-specific algorithmic innovations, a curated benchmark, and a knowledge base tailored to this field. We hope this context may be helpful when evaluating the paper's scope and contribution.
>
> For Q1, Q2, Q4:
>
> Compared with general-purpose multi-agent systems, HVR-Met introduces the following novelty:
>
> (1)Regarding the HVR mechanism, our framework advances the state-of-the-art in three key aspects.
>
> First, hypothesis generation is driven by anomaly detection rather than task descriptions [1, 2]. Our framework invokes the check\_anomaly tool to detect statistical extremes and identify the event type, then retrieves the corresponding diagnostic factors from the Guideline Library to formulate an initial physical hypothesis and its executable task chain. The content of each hypothesis factor is dictated by anomalous signals present in the data, not by open-ended exploration.
>
> Second, replanning is triggered when the computed indices or figures fail to exhibit the expected anomalies, rather than when task execution encounters errors, as is typical in general-purpose agents [3, 4].
>
> Third, the direction of replanning is constrained by domain knowledge rather than determined by generic reflection mechanisms [5, 6]. General-purpose agents typically resort to ''try a different approach'' or ''correct the previous step'' upon failure. By contrast, when HVR-Met rejects a hypothesis, it queries the Guideline Library for alternative diagnostic entries under the same weather type, ensuring that the replanning direction is guided by pre-defined domain knowledge rather than freely generated by the LLM.
>
> Together, these three aspects ensure that the HVR loop operates as a disciplined, domain-constrained reasoning process distinct from the generic plan-execute-retry paradigm.
>
> (2)Regarding the benchmark, we fill a gap in the evaluation landscape for extreme weather diagnostics. Existing meteorological AI evaluations focus predominantly on forecasting accuracy, lacking fine-grained assessment frameworks oriented toward the diagnostic reasoning process. We construct a comprehensive benchmark suite covering atomic-level subtasks, establishing a new evaluation standard for this field.
>
> (3)Regarding the knowledge base, we systematize scattered domain expertise into a structured, agent-callable knowledge repository. Through semi-automatic extraction of key information from a large corpus of extreme weather diagnostic papers, validated and supplemented by five senior forecasters, the resulting Guideline Library transforms implicit expert experience into explicit, retrievable domain priors for the agent workflow.
>
> Furthermore, regarding Question 3:
>
> The evaluation of the Figure Generation sub-task is grounded in independent, human-verified ground truth, rather than standards set by the VLM itself. Please refer to Section 4 and Figure 2 of our paper.
>
> For the evaluation of end-to-end diagnostic reports, we have increased the proportion of human-scored cases to 30\%.
>
> [1] Shen, Y., et al. "HuggingGPT: Solving AI Tasks with ChatGPT and its Friends in Hugging Face." Advances in Neural Information Processing Systems (2023).
>
> [2] Wei, J., et al. "Chain-of-Thought Prompting Elicits Reasoning in Large Language Models." Advances in Neural Information Processing Systems (2022).
>
> [3] Madaan, A., et al. "Self-Refine: Iterative Refinement with Self-Feedback." Advances in Neural Information Processing Systems (2023).
>
> [4] Shinn, N., et al. "Reflexion: Language Agents with Verbal Reinforcement Learning." Advances in Neural Information Processing Systems (2023).
>
> [5] Wang, G., et al. "Voyager: An Open-Ended Embodied Agent with Large Language Models." Advances in Neural Information Processing Systems (2023).
>
> [6] Hong, S., et al. "MetaGPT: Meta Programming for A Multi-Agent Collaborative Framework." International Conference on Learning Representations (2024).

---

> > ### Author Rebuttal · Reviewer_W4nN · 2026-04-03
> >
> > The novelty of this paper is not clear to me. I am not at all persuaded by their response, and I also feel that they ignored the substance of my review. I have decided to keep my score as Reject.

---

> > > ### Author Response · Authors · 2026-04-07
> > >
> > > The reviewer remarks that our previous response 'ignored the substance' of their review, yet does not specify the precise nature of this 'substance.' **In fact, we highly value specific feedback, as detailed suggestions are crucial for improving our manuscript.**
> > >
> > > We would like to respectfully clarify that our previous consolidation of Q1, Q2, and Q4 was guided by our understanding that the reviewer's underlying concern was the core innovation of HVR-Met.
> > >
> > > **As the novelty of HVR (Q1) was addressed previously, we focus below on clarifying Q2, Q3, and Q4.**
> > >
> > > **Q2:**
> > > Regarding the reviewer’s concern that "system's performance is heavily bound by the underlying capabilities of models”, we contend that the substantial gains provided by the HVR-Met framework and the meteorological knowledge base far exceed the baseline capabilities of foundation models. As clearly demonstrated by the ablation studies (Table 2 and Table 3), the removal of either the injected knowledge base or any core agent within HVR-Met results in a precipitous decline in the system's performance. This provides strong evidence that, in the absence of the structured reasoning paths and domain-specific constraints provided by HVR-Met, even the most advanced general large models will  ****fail to execute complex scientific diagnostic tasks.
> > >
> > > The reviewer suggests HVR-Met merely calls libraries like MetPy. However, our innovation lies not in reimplementing functions, but in the dynamic decision-making logic orchestrating them. Instead of a static pipeline, the HVR loop employs an anomaly-driven heuristic reasoning process: it formulates hypotheses, verifies multi-modal evidence, and dynamically replans. This goal-driven tool orchestration is a fundamentally novel paradigm in meteorological diagnosis.
> > >
> > > Despite the value of model fine-tuning and data representations, the operational deployment of extreme weather diagnosis is bottlenecked by interpretability and expert-level reasoning. HVR-Met addresses this by contributing a systematic integration of workflows, reasoning, and domain knowledge, rather than focusing on the foundation model.
> > >
> > > **Q3:**
> > >
> > > - *1. Evaluation Standards for the Figure Generation Sub-task are 100% Established by Human Experts*
> > >
> > > The reviewer's concern regarding limited human evaluation reflects a misunderstanding. While the 20% expert alignment (now 30% in the revision) applies exclusively to the 'End-to-End Diagnostic Assessment', the evaluation standards for the Figure Generation sub-task are entirely human-established. Specifically, 100% of the evaluation QA pairs—derived from 100 gold-standard journal images—were rigorously verified and corrected by five senior meteorologists.
> > >
> > > - *2. No Circular Validation Risks*
> > >
> > > The Figure Generation sub-task does not utilize the iterative HVR-loop. Figures produced by the Plotter and Image Checker are directly submitted to the Diagnostician (the evaluation VLM) along with the expert-corrected questions for binary evaluation.
> > >
> > > We respectfully address the reviewer's concern regarding VLM circularity: there is no overlap in their evaluation dimensions. The Image Checker focuses purely on "visualization quality" (meteorological plotting standards, isolines, legends), whereas the Diagnostician strictly evaluates the "identification of diagnostic signals" (causal reasoning for physical anomalies like vortices or shear lines).
> > >
> > > **Q4:**
> > >
> > > The reviewer misunderstands our architecture by conflating the System Prompt—which merely defines roles and constraints—with 'expert knowledge integration.' HVR-Met does not rely on a rigid template; rather, physical mechanisms are dynamically injected via RAG. Expert knowledge actually spans three layers: 1) an expert-verified Guideline Library, 2) an Index/Figure Knowledge Base (RAG for tools like MetPy/Cartopy), and 3) the System Prompt. Table 2 clearly shows that removing the Guideline Library drops the score from 3.94 to 0.40 with an unchanged System Prompt. This empirically confirms our system’s performance is driven by external knowledge bases, not the prompt itself.
> > >
> > > Furthermore, while the reviewer perceives "strict constraints and formatting rules" as a deficiency, in the context of multi-agent collaboration, precise role boundaries and operational constraints are fundamental prerequisites for ensuring system stability and reliability.
> > >
> > > Regarding generalizability, the system is proven across five extreme weather types (gale, rainstorm, snowstorm, coldwave, heatwave). Migrating to other Earth Science tasks requires no changes to the core HVR reasoning architecture or agents; only the external knowledge bases need replacing.
> > >
> > > **We sincerely thank the reviewer for their time and patience in reviewing our response. We hope these detailed clarifications comprehensively address all of your remaining concerns.**

---

### Official Review · Reviewer_LA3G · 2026-04-04

**Soundness:** 3
**Presentation:** 3
**Significance:** 3
**Originality:** 3
**Overall Recommendation:** 4
**Confidence:** 4

**Summary:**

This paper addresses the challenge of extreme weather diagnostics in forecasting systems, arguing that such scenarios require sophisticated multi-step reasoning, dynamic tool use, and expert-level prior knowledge. The authors identify key limitations in existing agent-based architectures, including insufficient integration of domain expertise, a lack of professional-grade iterative reasoning loops, and the absence of fine-grained validation frameworks for complex workflows under extreme conditions. To overcome these issues, they propose HVR-Meta, a multi-agent meteorological diagnostic system that deeply integrates expert knowledge through a “Hypothesis–Verification–Replanning” closed-loop mechanism designed to support iterative reasoning over anomalous meteorological signals. In addition, the paper introduces a novel benchmark centered on atomic-level subtasks to address gaps in current evaluation paradigms. Experimental results indicate that the proposed system performs effectively in complex extreme-weather diagnostic scenarios, suggesting its potential for improving reliability and structure in agent-based meteorological analysis.

**Compliance With Llm Reviewing Policy:**

Affirmed.

**Final Justification:**

weak accept

**Key Questions For Authors:**

1. How are extreme weather categories defined and partitioned in your benchmark? Are the categories based on established meteorological standards (e.g., WMO definitions), operational forecasting practices, or data-driven clustering? Additionally, how do you ensure coverage and balance across different types of extreme events?

2. In the reported experimental results, are the performance metrics based on pass@1, pass@k, or another aggregation strategy? If multiple attempts or iterative replanning steps are allowed within the HVR loop, how is this reflected in the final evaluation metric?

3. For the rubric-based report evaluation, how well do the judge model’s scores align with expert human meteorologists? Have you conducted inter-rater agreement analysis (e.g., correlation, Cohen’s kappa) between the LLM judge and human evaluators to validate reliability?

**Limitations:**

nan

**Strengths And Weaknesses:**

Strengths

1. The HVR-Met framework is clearly inspired by the professional “Weather Consultation” process and decomposes the diagnostic task into seven specialized agents with well-defined roles. This structured orchestration (e.g., Decomposer, Diagnostician, Data Specialist, Reporter) enhances interpretability and modularity, and reflects a thoughtful alignment between system architecture and real-world meteorological practice.

2. The introduction of the Hypothesis–Verification–Replanning (HVR) loop represents a principled attempt to enable iterative, abductive reasoning under complex extreme weather conditions. This design goes beyond simple task decomposition by incorporating dynamic feedback and replanning, which is particularly valuable for handling anomalous meteorological signals and uncertain diagnostic pathways.

3. The development of a comprehensive benchmark tailored to extreme weather diagnosis, especially with atomic-level subtask evaluation, is a meaningful contribution. Such fine-grained evaluation can provide clearer signals about agent capabilities and failure modes, and may serve as a useful foundation for future research in meteorological AI systems.

Weaknesses

1. While the paper introduces a comprehensive benchmark, it does not clearly specify how data quality is ensured. For example, it remains unclear how cases are selected, curated, validated, or annotated, and whether expert meteorologists are involved in quality control. Given the complexity and domain sensitivity of extreme weather diagnostics, rigorous data governance is critical.

2. In Table 1, the comparison includes GPT-5, Gemini-3-Pro, DeepSeek-R1, and Qwen3-Coder, which are strong and representative models. However, the absence of other competitive frontier models (e.g., Claude series) limits the comprehensiveness of the empirical comparison. A broader selection would strengthen claims about relative performance and generalizability.

3. The evaluation of diagnostic reports relies on rubric-based scoring, presumably using LLM judges. However, the paper does not analyze whether different judge models yield consistent scores under the same rubric. Without robustness checks across multiple evaluators, the reliability and objectivity of the report-level assessment may be questioned.

---

### Decision · Program_Chairs · 2026-04-30

**Decision:**

Accept (regular)

**Comment:**

The paper studies extreme weather diagnostics task with an agentic system. Authors formulate the task in an agentic manner, propose a ``Hypothesis-Verification-Replanning'' closed-loop mechanism, and conduct experiments with diverse foundation models. While there are concerns that the paper does not introduce novel machine learning algorithm, others reviewer think this is a useful application paper. The AC agrees that defining new tasks is valuable for the AI community and the work is also insightful for the climate domain.